# Bias-blind and bias-aware assimilation of leaf area index into the Noah-MP land surface model over Europe

Samuel Scherrer[1,2], Gabriëlle De Lannoy[2], Zdenko Heyvaert[1,2], Michel Bechtold[2], Clement Albergel[3], Tarek S. El-Madany[4], and Wouter Dorigo[1]

[1]Department of Geodesy and Geoinformation, TU Wien, Vienna, Austria
[2]Department of Earth and Environmental Sciences, KU Leuven, Heverlee, Belgium
[3]European Space Agency Climate Office, ECSAT, Didcot, UK
[4]Max Planck Institute for Biogeochemistry, Department Biogeochemical Integration, Jena, Germany

**Correspondence:** Samuel Scherrer (samuel.scherrer@geo.tuwien.ac.at)

**Abstract.** Data assimilation (DA) of remotely sensed leaf area index (LAI) can help to improve land surface model estimates of energy, water, and carbon variables. So far, most studies have used bias-blind LAI DA approaches, i.e. without correcting for biases between model forecasts and observations. This might hamper the performance of the DA algorithms in the case of large biases in either observations or simulations, or both. We perform bias-blind and bias-aware DA of the Copernicus Global
Land Service LAI into the Noah-MP land surface model forced by the ERA5 reanalysis over Europe in the 2002–2019 period, and evaluate how the choice of bias correction affects estimates of gross primary productivity (GPP), evapotranspiration (ET), runoff, and soil moisture.

    In areas with a large LAI bias, the bias-blind LAI DA leads to a reduced bias between observed and modelled LAI, an improved agreement of GPP, ET, and runoff estimates with independent products, but a worse agreement of soil moisture
estimates with the European Space Agency Climate Change Initiative (ESA CCI) soil moisture product. While comparisons to in-situ soil moisture in areas with weak bias indicate an improvement of the representation of soil moisture climatology, bias-blind LAI DA can lead to unrealistic shifts in soil moisture climatology in areas with strong bias. For example, when the assimilated LAI data in irrigated areas are much higher than those simulated without any irrigation activated, LAI will be increased and soil moisture will be depleted. Furthermore, the bias-blind LAI DA produces a pronounced sawtooth pattern due
to model drift between DA updates, because each update pushes the Noah-MP leaf model to an unstable state. This model drift also propagates to short-term estimates of GPP and ET, and to internal DA diagnostics that indicate a suboptimal DA system performance.

    The bias-aware approaches based on a priori rescaling of LAI observations to the model climatology avoid the negative effects of the bias-blind assimilation. They retain the improvements of GPP anomalies from the bias-blind DA, but forego
improvements in the root mean square deviation (RMSD) of GPP, ET, and runoff. As an alternative to rescaling, we discuss the implications of our results for model calibration or joint parameter and state update DA, which has the potential to combine bias reduction with optimal DA system performance.

# 1 Introduction

Vegetation plays a major role in climatic interactions between the land surface and the atmosphere. Via transpiration and
photosynthesis, it contributes to the exchange of energy, water, and carbon at the surface, and links the moisture in the deeper
soil layers to the atmosphere (Bonan, 2019). On short timescales, these exchanges can impact precipitation and atmospheric
circulation (Betts et al., 1996; Miralles et al., 2016). On longer timescales, the net uptake of $CO_2$ by vegetation (Friedlingstein
et al., 2022) might be decreased due to climate change, contributing to rising $CO_2$ levels (Green et al., 2019; Walker et al.,
2021). Land surface models (LSMs) are often used to estimate these exchange fluxes as part of Earth system models or as land
component in numerical weather prediction (NWP) systems (e.g., Balsamo et al., 2009; Lawrence et al., 2019; Skamarock
et al., 2019). An accurate description of vegetation in LSMs can therefore improve estimates of evapotranspiration (ET) in
NWP (Boussetta et al., 2013), or can be used to estimate how vegetation will develop under a changed climate (Laanaia et al.,
2016) and how this affects the land carbon sink (Tharammal et al., 2019a, b; Green et al., 2019).

However, the dynamic simulation of vegetation in global LSMs is still in its infancy and has large uncertainties, especially
in dry climates (Fox et al., 2018; Mahmud et al., 2021). Satellite-based vegetation data assimilation (DA) can be used to
reduce the uncertainties of the vegetation-related LSM estimates. Satellite-derived leaf area index (LAI) is commonly used for
DA, because it can be derived from optical sensors fairly accurately (Fang et al., 2019) and is also available as model state
variable in several land surface models with a dynamic vegetation component. Satellite LAI has for example been assimilated
into the *Interactions between Soil Biosphere Atmosphere* (ISBA) LSM (Sabater et al., 2008; Barbu et al., 2014; Fairbairn
et al., 2017; Albergel et al., 2017; Mucia et al., 2020), the *Noah LSM with multiparameterisation options* (Noah-MP; e.g.,
Kumar et al., 2019b, 2021; Rahman et al., 2022b; Nie et al., 2022), the *Community Land Model* (CLM; e.g., Fox et al., 2018;
Ling et al., 2019), and the *Carbon-Tiled ECMWF Scheme for Surface Exchange over Land* (CTESSEL; e.g., Jarlan et al.,
2008). Alternatives are, for example, to use microwave brightness temperatures to simultaneously update soil moisture and
LAI (Sawada and Koike, 2014; Sawada et al., 2015) or to use microwave vegetation optical depth (VOD) retrievals to update
LAI (Kumar et al., 2020, 2021).

The most commonly used methods for assimilating LAI into LSMs are based on the Kalman filter. A fundamental assumption
of these methods is that modelled and observed LAI are unbiased. Yet, in reality, biases nearly always exist. This includes
biases of both model estimates and observations with respect to the unknown true value, and between the model estimates and
observations themselves. If the observations are closer to the true value than the model estimates, a "bias-blind" DA (Dee,
2005) is able to correct the model bias to some extent, because it pulls the model closer towards the observations and, hence,
the true values. This comes at the risk of introducing unintended negative side effects. For example, it is possible that other
processes (e.g., transpiration) are only represented well for a biased model climatology. Large updates in a subset of the model
state might therefore propagate to other model components, which can negatively affect estimates of state variables and fluxes
of these processes (De Lannoy et al., 2007b; Crow et al., 2020). Furthermore, if the model equilibrium state is far away from
the observations, the updates towards the observations might not persist for long. Instead, the model drifts back towards its
original state, leading to a sawtooth-like pattern in the resulting time series and potentially also to unrealistic water, carbon and

energy flux estimates (Dee, 2005; De Lannoy et al., 2007b). Changes in observation frequency or periodically missing data may then also introduce spurious trends in the analysis (Dee, 2005).

Most LAI assimilation studies so far used bias-blind approaches, i.e. they did not apply any bias correction methods to account for existing biases between modelled LAI and observed LAI. This is often justified by the argument that the bias is caused by model deficiencies (e.g., Fairbairn et al., 2017; Fox et al., 2018; Albergel et al., 2020). Nonetheless, there are indications that the presence of bias affects the performance of LAI assimilation. Albergel et al. (2017) and Albergel et al. (2020) noticed systematic drifts towards the previous model estimate on days without observations. Kumar et al. (2019b); Mocko et al. (2021) also found model drifts leading to sawtooth patterns in the analysed LAI when using the Noah-MP LSM with dynamic vegetation.

Various techniques have been used to limit the negative effects listed above. Albergel et al. (2017, 2020) and Mucia et al. (2021) additionally assimilated surface soil moisture retrievals. This additional constraint can help to prevent negative side-effects of the LAI DA on the model hydrology, but only in regions and periods where sufficient soil moisture observations are available. Kumar et al. (2019b); Mocko et al. (2021) and Rahman et al. (2022b) interpolated their assimilated LAI product to daily values to prevent issues due to different observation frequencies and to limit the drift towards the original equilibrium state. Fox et al. (2018) adaptively inflated the model error in case of large bias between modelled LAI and the observations. The latter two techniques force the analysis to stay close to the observations, which begs the question of whether it might be more suitable to use a direct insertion approach or to prescribe the observed LAI instead of modelling it dynamically, as for example done by Maertens et al. (2021); Huang et al. (2022).

Bias-aware data assimilation is another possible avenue to handle bias between models and observations. This includes a priori rescaling approaches, which map the observations into the model space based on a priori estimates of model and observation statistics (e.g., Reichle and Koster, 2004; Jarlan et al., 2008; Khaki et al., 2020), or online approaches which adaptively estimate dynamic bias corrections (e.g., Derber and Wu, 1998; Dee, 2005; De Lannoy et al., 2007a). Only a few studies considered bias-aware approaches based on rescaling for LAI DA (Jarlan et al., 2008; Khaki et al., 2020). However, no study so far directly compared bias-blind and bias-aware LAI DA.

In this article, we compare the bias-blind LAI DA with bias-aware LAI DA using two a priori rescaling techniques commonly used for satellite DA. More specifically, we assimilate Copernicus Global Land Service (CGLS) LAI (Smets et al., 2019) into the Noah-MP model (Niu et al., 2011) forced with the fifth-generation European Center for Medium-Range Weather Forecasts (ECMWF) Reanalysis (ERA5; Hersbach et al., 2020) reanalysis over Europe, and quantify the effect of bias-blind and bias-aware DA on vegetation and surface water flux and state estimates.

A detailed description of the used model, data, and rescaling approaches can be found in section 2. Section 3 shows the impacts of the bias-blind DA on the vegetation and hydrology model estimates, evaluates the results using independent reference datasets, and compares the model simulations to in situ data from Majadas, Spain. Additionally, we provide an analysis of the sawtooth pattern in the bias-blind DA and of internal DA diagnostics. We discuss the implications of our results for LAI DA design and model calibration in section 4. A summary of our main conclusions is given in section 5.

## 2 Data & Methods

### 2.1 Land surface model

We used the Noah-MP LSM (Niu et al., 2011; Yang et al., 2011) version 4.0.1 with dynamic vegetation as implemented in the NASA Land Information System (LIS; Kumar et al., 2006; Peters-Lidard et al., 2007)). The Noah-MP LSM is based on the Noah LSM, which is widely used for land surface modelling and DA on a regional to global scale (e.g., Rodell et al., 2004; Kumar et al., 2014, 2019a; Maertens et al., 2021). Noah-MP includes a multitude of optional improvements for snow, water, and vegetation modelling. It has already been used to update LAI using optical satellite imagery (Kumar et al., 2019b; Erlingis et al., 2021; Rahman et al., 2022b) and microwave vegetation optical depth (Kumar et al., 2020, 2021).

The dynamic vegetation model of Noah-MP is based on the vegetation model in the Biosphere–Atmosphere Transfer Scheme (BATS) model (Dickinson et al., 1998). In this model, gross primary productivity (GPP) is allocated to the four vegetation carbon pools (leaves, non-woody stems, wood, and fine roots) in each simulation step. LAI is calculated from leaf carbon mass by multiplying with a vegetation type dependent specific leaf area. It can feed back to other model state variables and fluxes via its effect on photosynthesis, evapotranspiration (ET), precipitation interception, and runoff. Changes in LAI can therefore also induce changes in the model hydrology. A more detailed overview of the dynamic leaf model in Noah-MP is given in Appendix A.

Maps of soil texture and land cover, and multiple parameters based on these, are required as input to the model and were taken from the NCCS Dataportal (https://portal.nccs.nasa.gov/lisdata_pub/data/PARAMETERS/; Tian et al., 2008). We used the STATSGO-FAO (*State Soil Geography - Food Agricultural Organisation*) soil texture map produced by the *National Center for Atmospheric Research* (NCAR). For vegetation, we used the IGBP-NCEP (*International Geosphere-Biosphere Programme - National Centers for Environmental Prediction*) land cover map based on Friedl et al. (2002). This map classifies some pixels in France, Spain, Ireland and Germany as evergreen broadleaf forests, which the model interprets as tropical rainforests. We therefore replaced these pixels with the land cover class from the University of Maryland (UMD) land cover map (Hansen et al., 2000). The soil texture and land cover maps are available on a 0.01° regular grid and were upscaled to a 0.25° grid using the largest fraction within a model grid cell.

As forcing, Noah-MP requires the lowest level atmospheric model (about 10 m above ground level) air temperature, wind speed, specific humidity and pressure, the downwelling fluxes of shortwave and longwave radiations, as well as precipitation (partitioned into solid and liquid phases). We used data from ERA5, the latest ECMWF reanalysis, for this purpose. The ERA5 forcings have an original resolution of 31 km and were mapped to a 0.25° regular grid. The initial model state was obtained from a 30-year deterministic spinup run, cycling 3 times with the forcing data from 2000 to 2010, followed by 2 years of ensemble spinup from 2000 to 2002.

The model domain in this study covers Europe, as well as parts of Northern Africa and the Middle East on a regular grid at a 0.25° resolution (ranging from 29.875°N, −11.375°E to 71.625°N, 40.125°E). It includes a wide range of climates and vegetation types, from tundra and boreal forests in Scandinavia to the Sahara Desert. We performed the model simulations from 2002 through 2019, using a 15-minute simulation time step and outputting daily averages centred at 0:00 UTC.

## 2.2 LAI observations

We assimilated the Copernicus Global Land Service (CGLS) satellite LAI product version 2 derived from *Project for On-Board Autonomy - Vegetation* (PROBA-V) and *Satellite Pour l'Observation de la Terre - Vegetation* (SPOT-VGT) (Verger et al., 2014). This product has been used for LAI DA before, e.g. by Barbu et al. (2014), Albergel et al. (2017), and Mucia et al. (2020). The 1 km resolution CGLS LAI product is provided as 10-daily images composed from an adaptive window of 15 to 60 days, depending on the availability of valid measurements (Smets et al., 2019). We masked out gap-filled values and upscaled the data to 0.25° resolution by averaging over all observations within one model grid cell. In contrast to Kumar et al. (2019b), we did not interpolate the LAI to daily values. This way, we (i) do not introduce observation error auto-correlations, (ii) allow our results to be generalizable to LAI data sets (or proxy data sets as used in Kumar et al. (2020); Mucia et al. (2021)) with less frequent observations or changes in observation frequency, and (iii) can investigate if the filter efficiently interpolates and operates as intended (or assumed). We assimilated the aggregated data every 10 days at 0:00 UTC, where and when they are available.

## 2.3 Data assimilation

We used a one-dimensional ensemble Kalman Filter (EnKF; Evensen, 2003) for assimilating the CGLS LAI observations into the Noah-MP LSM. The EnKF is a two-step procedure. First, the model simulates the land surface state $x^f(t)$ at the next assimilation time step $t$ (forecast). Then, the model state is updated to agree better with the observations $y(t)$, resulting in the analysis $x^a(t)$. The magnitude of the update (increment) depends on the innovations (observation minus forecast) and the relative sizes of the forecast and observation error variances. In a properly configured DA system, the normalised innovations (innovations divided by total error standard deviation) should be temporally uncorrelated and follow a standard normal distribution, i.e., the innovation sequence should be a white noise sequence with zero mean and unit standard deviation (Desroziers et al., 2005). Following prior work on LAI DA with Noah-MP (Kumar et al., 2019b; Mocko et al., 2021; Rahman et al., 2022b, a), we use the EnKF to update the model LAI, i.e., the state vector consists only of LAI. It is chosen to not add soil moisture in the updated state vector, for reasons discussed in section 4.4.

In the EnKF, the forecast error is estimated based on an ensemble of model simulations. We used 24 ensemble members, one of which was driven by the original forcing data, while the others were driven by perturbed radiation and precipitation forcing data. Additionally, we applied normally distributed perturbations to the model LAI state variable with a mean of zero and a standard deviation of $0.01\,\mathrm{m^2m^{-2}}$ every 3 hours for the 23 perturbed ensemble members. The unperturbed ensemble member was used to correct for perturbation biases due to nonlinear processes using the method described by Ryu et al. (2009). All of the perturbation specifications and the observation error standard deviation of $0.05\,\mathrm{m^2m^{-2}}$ were set following Kumar et al. (2019b).

To remove systematic differences between the modelled and observed LAI, we implemented either one of two a priori rescaling methods: climatological cumulative distribution function (CDF) matching and a seasonal rescaling of the first and second moments. CDF-matching is commonly used for soil moisture DA without distinguishing the various seasons (e.g.,

Reichle and Koster, 2004; Drusch et al., 2005; Draper et al., 2012; Parrens et al., 2014; Barbu et al., 2014). It attempts to correct the biases in all statistical moments by non-linearly transforming the observation data such that the empirical CDF of the rescaled LAI data matches the empirical CDF of the modelled data. To estimate the empirical CDFs for each grid cell individually in a robust way, we opted to bin the data between the $2^{nd}$ and $98^{th}$ percentile. We then estimated the CDF by linearly interpolating the percentile values between the bin edges. For values outside the [2, 98] interval, the lines for the first and last bin are extrapolated to 0 and 100, respectively. The resulting curve is discretised into 100 equally spaced bins over the full data range for use in the numerical rescaling procedure. When using the CDF-matching for rescaling, the observation error standard deviations are also rescaled for each grid cell individually by multiplying with the ratio of the modelled and observed LAI standard deviations.

The seasonal rescaling method is an adaption of the additive seasonal mean correction scheme commonly used for brightness temperature DA (De Lannoy and Reichle, 2016; Lievens et al., 2017; Girotto et al., 2019; Bechtold et al., 2020). Similar to LAI, brightness temperatures also have a strong seasonal component. The additive rescaling only corrects biases in the first moment (mean). This is valid if the difference in anomaly variance between the model and observations is related to different error levels, i.e., the signal variances are similar (Yilmaz and Crow, 2013). In our case, differences in anomaly variance are strongly driven by differences in the dynamic range of observations and model estimates. We assume that the differences in the dynamic range also result in differences in error levels and therefore additionally corrected for the ratio of the standard deviation of model and observation.

For the seasonal rescaling, we calculated the rescaled observation values $LAI_o'$ at each time $t$ via

$$LAI_o'(t) = \mu_m(doy(t)) + \frac{\sigma_m}{\sigma_o} \cdot (LAI_o(t) - \mu_o(doy(t))),$$

with $\mu_\star(doy(t))$ the mean modelled ($m$) or observed ($o$) LAI value for the given day of year, and $\sigma_\star$ the standard deviation of the modelled or observed LAI time series at individual grid cells. The latter is mainly indicative of the magnitude of the seasonal variations. The mean seasonal cycle of modelled and observed LAI was estimated through a three-step procedure as implemented in the python package *pytesmo* (Paulik et al., 2022), i.e. (i) apply a smoothing with a 5-day moving window, (ii) average values over days of year across multiple years ($doy$), and (iii) smooth the obtained seasonal cycle using a window of 31 days. When using the seasonal rescaling we also rescale the observation error standard deviation for each grid cell individually by multiplying with $\sigma_m/\sigma_o$.

We performed four model runs in total, one open loop (OL) run without any data assimilation (but applying the same perturbations), and one bias-blind and two bias-aware LAI DA runs:

- no bias correction (*bias-blind*)

- CDF matching for bias correction (*CDF-matched*)

- seasonal bias correction (*seasonally scaled*)

## 2.4 Evaluation metrics

To evaluate the performance of the OL and DA simulations, we calculated the root mean square deviation (RMSD), linear correlation ($R$) and linear anomaly correlation ($R_{anom}$) with independent reference datasets.

RMSD is a common measure for the overall disagreement between two datasets. It consists of a bias component due to bias in the first and second moments (mean and variance bias) and a correlation component due to disagreement of the temporal patterns (Gruber et al., 2020). When applied to time series with a strong seasonal cycle, as is the case for most variables we evaluate, it is dominated by mean bias and bias in the representation of the seasonal cycle. It is therefore mainly indicative of systematic disagreement between modelled and reference data.

Linear correlation $R$ is not affected by mean or variance bias, but in the case of a strong seasonal cycle, it is also dominated by bias in the representation of the seasonal cycle. It therefore quantifies how well the shapes of the seasonal cycles (e.g., peak location, phase shift) of two datasets match.

For assessing the agreement in the intra- and inter-annual temporal variations, we used linear anomaly correlation ($R_{anom}$). The anomalies are calculated by subtracting the long-term mean seasonal cycle for the 2003–2019 period from the original data for each grid cell. The mean seasonal cycle is calculated the same way as the seasonal cycle used for the seasonal observation rescaling (see subsection 2.3).

To make the metric improvements comparable over different variables and metrics we calculated the normalised information contributions (NIC; Kumar et al., 2009, 2014) for the three metrics:

$$NIC\ RMSD = \frac{RMSD_{OL} - RMSD_{DA}}{RMSD_{OL}}$$

$$NIC\ R = \frac{R_{DA} - R_{OL}}{1 - R_{OL}}$$

$$NIC\ R_{anom} = \frac{R_{anom,DA} - R_{anom,OL}}{1 - R_{anom,OL}}.$$

Positive NIC values indicate an improvement compared to the OL run (up to a maximum of 1), negative NIC values indicate a deterioration compared to the OL run.

## 2.5 Reference data

We used a range of reference data for assessing the impact of the different DA methods on different simulated variables. The vegetation and carbon cycle representations were evaluated via GPP, whereas the hydrological component was evaluated via evapotranspiration (ET), soil moisture (SM), and runoff, either using in situ data or as spatially gridded satellite-based products. We mapped all reference data to the model grid ($0.25°$) by averaging (for gridded datasets) or nearest neighbour matching (for in situ data). Where available, evaluations were performed using daily model output. Otherwise, we averaged the model output to the temporal resolution of the reference product. In the bias-blind DA, some variables contained strong trends in the first DA year (2002), caused by the induced climatology changes. We therefore limited the evaluation to 2003–2019.

### 2.5.1 FluxSat GPP

FluxSat (Joiner and Yoshida, 2021) provides global daily estimates of GPP retrieved from the *Moderate Resolution Imaging Spectroradiometer* (MODIS). The retrieval is based on an empirical light use efficiency model that estimates GPP via an artificial neural network (ANN) approach. The ANN was trained using in situ estimates of GPP from eddy covariance towers (FLUXNET). FluxSat agrees well with independent eddy covariance tower measurements (Joiner and Yoshida, 2020) and has been shown to outperform other GPP retrieval approaches (Joiner et al., 2018). Since the GPP estimates of FluxSat are based on data from optical sensors (although different from the ones used in our study), they might not be fully independent of the assimilated LAI observations, and especially correlation metrics might overestimate the DA skill improvements.

### 2.5.2 SIF

Sun-induced fluorescence (SIF) is a direct measure of photosynthetic activity and is mostly linearly correlated to GPP (Frankenberg et al., 2011) and ET (Maes et al., 2020). It is commonly used to evaluate improvements in the representation of GPP due to LAI data assimilation (Leroux et al., 2018; Kumar et al., 2019b; Albergel et al., 2020). We used a fused dataset from the *SCanning Imaging Absorption SpectroMeter for Atmospheric CHartographY* (SCIAMACHY) and the *Global Ozone Monitoring Experiment-2* (GOME-2) (Wen et al., 2021), which provides monthly global SIF estimates at a $0.05°$ resolution. Hence, the comparison with OL and DA runs was performed on monthly averages of modelled GPP. In contrast to FluxSat GPP, SIF is independent of the assimilated LAI observations, since it uses a different retrieval approach. Under extreme conditions, the linear relationship of SIF and GPP can break down (Martini et al., 2022). Therefore, similarly to FluxSat GPP, evaluations against SIF should be analysed carefully. Since we do not explicitly model SIF but only use it as a GPP proxy, we evaluated it only in terms of $R$ and $R_{anom}$.

### 2.5.3 GLEAM ET

The Global Land Evaporation Amsterdam Model v3 (GLEAM; Martens et al., 2017; Miralles et al., 2011) ET dataset is a gridded ET product based on a land surface model and satellite observations. It has been evaluated against other products in various benchmarking activities (Greve et al., 2014; Martens et al., 2016, 2017, 2018), and it has been used for assessing DA systems (e.g., Albergel et al., 2019; Bonan et al., 2020; Kumar et al., 2019b; Rahman et al., 2022b, a). We used version 3.6b, as it provides data in our evaluation period (2003-2019) and does not rely on either reanalysis as forcing data or optical data for dynamic inputs. It is thus largely independent of the assimilated CGLS LAI and of the Noah-MP-modelled ET, but inevitably suffers from model assumptions and input errors.

GLEAM calculates ET as a combination of potential evaporation (based on the Priestley-Taylor equation), stress, and interception (based on the Gash model). Water stress is based on a soil moisture model included in GLEAM, and an additional scaling based on observations of vegetation optical depth, a proxy for vegetation water content.

Since the soil moisture model does not include irrigation explicitly, it will provide biased estimates over strongly irrigated areas (Chen et al., 2021; Shah et al., 2019). Evaluations of absolute values (e.g., via RMSD) over irrigated areas should

therefore be analysed carefully, as they might show decreased performance stemming from an actually improved representation of irrigation (as for example in Thiery et al., 2017), even if satellite-based soil moisture anomalies were assimilated and might partly compensate for missed irrigation.

### 2.5.4 ESA CCI soil moisture

The *European Space Agency* (ESA) *Climate Change Initiative* (CCI) soil moisture (SM) v07.1 (Dorigo et al., 2017) dataset is a merged product combining soil moisture retrievals from a multitude of satellites. We use the COMBINED product, which includes soil moisture from passive satellites retrieved with the Land Parameter Retrieval Model (LPRM; Owe et al., 2008), and soil saturation from active satellites retrieved with the TU Wien change detection method (Wagner et al., 1999; Naeimi et al., 2009).

The merging is based on a variance-weighted average, with error variances obtained from a triple collocation error characterisation (Gruber et al., 2019). Recent releases also include a homogenisation of breaks that may be introduced during the merging (Preimesberger et al., 2020). The merging process also uses soil moisture estimates from the *Global Land Data Assimilation System* (GLDAS; Rodell et al., 2004) as a scaling reference, and the climatology of the final product is therefore the climatology of GLDAS. As such, we performed comparisons to ESA CCI SM only in terms of anomaly correlations with the uppermost soil moisture layer simulated by Noah-MP (0-10 cm).

### 2.5.5 ISMN soil moisture

The International Soil Moisture Network (ISMN; Dorigo et al., 2021, 2011, 2013) provides in situ soil moisture data from over 70 soil moisture sensor networks around the globe. We calculated daily averages of in situ soil moisture data from the depths $0\,cm$ to $10\,cm$ (SM1) and $10\,cm$ to $40\,cm$ (SM2) from all networks providing station data within our modelling domain (see Table C1). Only data with quality flag "good" have been used, and we discarded stations with less than 1000 days of valid data within our evaluation period. Metrics were computed based on a nearest neighbour matching between ISMN stations and model grid coordinates. In the case of multiple stations per model grid cell, we averaged the metrics of these stations to obtain a single value per model grid cell. Since soil moisture climatology and absolute values strongly depend on sub-grid scale factors like slope and soil texture, we only compared the in situ values in terms of anomaly correlation $R_{anom}$. Additionally, we restrict the comparison to ISMN stations that have been shown to be representative at the $0.25°$ resolution with a triple collocation analysis involving ISMN, ERA5-Land layer 1 SM, and ESA CCI SM (Dorigo et al., 2021, Fig. 7).

### 2.5.6 GRDC runoff

To evaluate the effects of the assimilation on modelled runoff, we used monthly river discharge station data from the Global Runoff Data Centre (GRDC; Koblenz, Germany). The station basins were derived from the provided watershed boundaries (GRDC, 2011).

The comparison of modelled total (surface + subsurface) runoff to station river discharge followed the approach of Koster et al. (2014) and Koster et al. (2018), who compared river discharge with 10-daily basin-averaged runoff. We restricted the analysis to 271 stations in Europe with a record of more than 10 years and a basin area between $625\,\mathrm{km}^2$ and $100{,}000\,\mathrm{km}^2$. The lower bound follows Kumar et al. (2014), the upper bound was increased compared to Kumar et al. (2014) and Koster et al.

(2018) in order to have more available stations in southern Europe (mainly Spain). We account for the larger area by using monthly averages instead of the 10-daily averages that were used by Koster et al. (2018). Basins with a Pearson correlation of less than 0.4 with respect to the OL run were excluded, so that the evaluation was not hampered by basins that are likely strongly affected by unmodelled processes (e.g., damming or irrigation).

### 2.5.7 Site data from Majadas

The ecosystem research site Majadas de Tiétar (Casals et al., 2009) is located in the center of the Iberian Peninsula at $39°56'25''$N $5°46'29''$W and categorised as a semi-arid savanna type ecosystem (El-Madany et al., 2018) with a canopy height of $8.7 \pm 1.25\,\mathrm{m}$, and a fractional canopy cover is $23.0 \pm 5.3\%$ (Bogdanovich et al., 2021). In the land cover map used in the model, the grid cell containing the research site is classified as "savanna". The accumulated annual precipitation at the site is about $650\,\mathrm{mm}$ with a large inter-annual variability. The mean LAI at the site changes strongly throughout the year between

$0.55 — 2.15\,\mathrm{m}^2\mathrm{m}^{-2}$ with lowest values during summer and highest values during late spring. The soil is an Abruptic Luvisol with a sandy upper layer (Nair et al., 2019). In the model, the grid cell containing the research site uses parameters for a loamy sand texture.

The research site consists of three eddy covariance towers with non-overlapping footprints climatologies and similar instrumental setups (El-Madany et al., 2021). For this analysis, the data of the tower with the FLUXNET ID ES-LM1 are used. A

detailed description of the instrumental setup and data processing can be found in El-Madany et al. (2018, 2021). In short, the soil moisture data are collected with four profile probes enviroSCAN (Sentek) measuring at 10, 20, 30, 50 and 100 cm plus an ML3 (Delta T) sensor at 5 cm close to each profile probe. The soil moisture data were further aggregated to depth levels representing the Noah-MP soil moisture layers for each of the 4 profiles.

Eddy covariance data were collected at $20\,\mathrm{Hz}$ with a R3-50 (Gill) and a LI-7200 $CO_2$ and $H2O$ gas analyser (Licor Bio-

science) at $15\,\mathrm{m}$ above ground. Raw data were processed with EddyPro (Fratini and Mauder, 2014) to calculate fluxes of ET and $CO2$ at half hourly intervals. Subsequently, u*-threshold estimation, gap-filling and flux partitioning were applied using REddyProc (Wutzler et al., 2018). The resulting continuous time-series of ET and GPP were aggregated together with other meteorological parameters to hourly timestamps, from which daily averages were computed.

### 2.6 Analysis of Noah-MP equilibrium LAI

As will be seen later, each update step in the bias-blind DA is followed by a strong drift of the model LAI towards the earlier forecast values, i.e. the bias-blind DA system quickly "forgets" systematic corrections made in earlier steps. This indicates that there is a stable equilibrium LAI (i.e. a model-based 'attractor') whose value is not modified by the bias-blind LAI DA. To

make full use of the information contained in the observations, a bias-blind DA system should also modify this equilibrium LAI value to have more persistent DA updates.

An analysis of the Noah-MP leaf growth model (Appendix A) shows that the main factors influencing the equilibrium LAI value are (i) root zone soil moisture, represented via the soil moisture factor $\beta$, and (ii) leaf parameters, e.g. specific leaf area (SLA, leaf mass per area). Including $\beta$ or SLA in the DA state vector could thus help to obtain more persistent updates.

We therefore analysed how sensitive the equilibrium LAI is to these variables using a climatological approximation of the Noah-MP leaf model (shown in Appendix B). The result of this analysis is presented in subsection 2.6 for two example sites

with constrasting bias between Noah-MP and CGLS, (i) the Majadas site in Spain, where observed LAI is much lower than modelled LAI, and (ii) the Nile delta, where observed LAI is much higher than modelled LAI.

## 2.7 Evaluation of short-term DA effects

To evaluate how biased updates affect the short-term model performance, we analyse day-to-day differences of model states and fluxes. In the OL, the day-to-day differences are driven by day-to-day variations in the forcing. If averaged over larger areas

or multiple years, this corresponds mainly to the fluctuations of the mean seasonal cycle. For LAI, GPP, and ET, which are high in summer and low in winter, we therefore expect positive day-to-day differences in spring, corresponding to leaf growth and increase in GPP and ET, and negative day-to-day differences in autumn, corresponding to leaf shedding and decrease in GPP and ET.

Large update steps in the bias-blind DA can induce model instabilities. In this case, the subsequent day-to-day differences

are strongly impacted by the unstable artificial response to the update step instead of reacting to the physical forcing input.

To detect if such model instabilities occur, and to what extent they propagate to flux estimates of the model, we evaluated differences in the estimates between day 2 and day 1 after assimilation, as well between day 1 and day 2 before assimilation (the latter can also be interpreted as approximately day 9 minus day 8 after after assimilation). A comparison of these also gives an indication of how long the DA-induced effects persist. For each pixel and month, we calculated the median of these

day-to-day differences over all years from 2003 to 2019 and normalised it with the monthly standard deviation of the variable values over the same multi-year time range (as a measure of the local within-month variation).

## 3 Results

### 3.1 Mean impact of bias-blind DA

Figure 1 compares mean values of OL and bias-blind DA results (relative to mean OL values) for different variables, for the

months of April through October across 17 years (2002–2019). The bias-blind DA decreases growing-season LAI over large parts of the domain or has a neutral impact. It only increases in the Alps and the Scandinavian Mountains. The regions with a large change in mean LAI are mostly semi-arid and include the Iberian Peninsula, Northern Africa, the Middle East, Turkey, and Ukraine, where modelled LAI is much higher than observed LAI, and modelled LAI is therefore strongly decreased by the

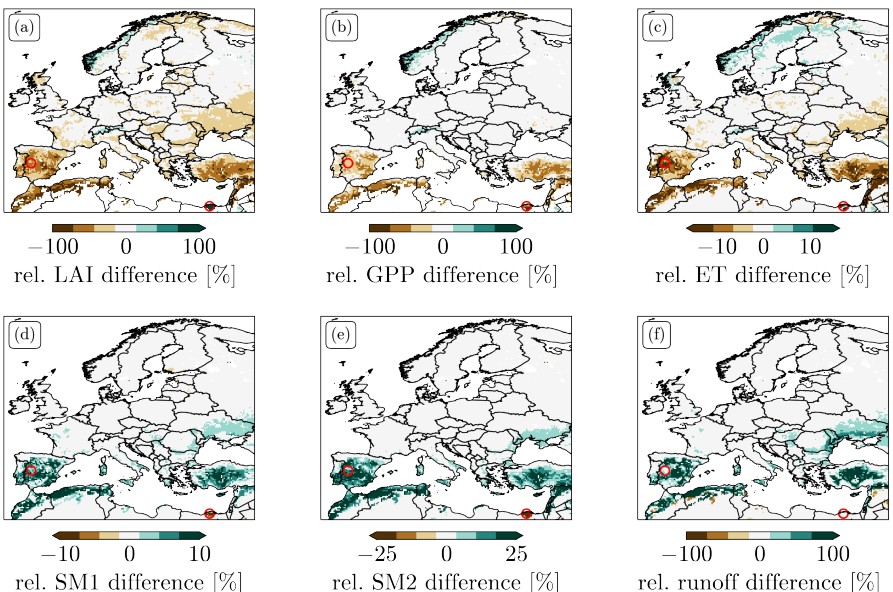

**Figure 1.** Relative differences between temporal mean values of OL run and bias-blind DA run for (a) LAI, (b) GPP, (c) ET, (d) SM1 (0–10 cm), (e) SM2 (10–40 cm), and (f) runoff, for the months of April through October 2002–2019. **Note the different colourbar ranges**. The red circles mark the example locations studied below, i.e., Majadas in Spain, and the Nile delta region in Egypt.

bias-blind DA. In contrast, LAI increases in the Nile delta, because the lack of irrigation forcing limits the model's ability to
grow vegetation.

Differences in mean GPP show similar patterns, but with a weaker impact overall, especially in Central and Eastern Europe. One exception is the Nile delta, where growing-season GPP decreases while LAI increases. Relative differences in mean ET are much lower (note the different colour bar range), but with similar large-scale patterns as for GPP. On the Iberian Peninsula, the patterns differ slightly: the largest relative differences are in the Western part, mainly over the Duero and Tajo basins. Over
Scandinavia, ET increases, except for the northernmost parts.

ET links the vegetation model to the hydrology model; consequently, the LAI assimilation also affects soil moisture and runoff. A reduction in LAI and hence transpiration leads to a reduction in soil moisture depletion. The effect is larger on deeper soil moisture layers than on surface soil moisture since the deeper layers are more strongly coupled with transpiration. In regions where LAI is strongly reduced by the DA, the relative increase in mean SM2 is about 20%. For runoff, the relative
increase even reaches 100%. In contrast, in the Nile delta, the increase in LAI leads to a reduction in soil moisture via transpiration. In the Alps and Scandinavia, soil moisture is not affected systematically, since the water balance is dominated by runoff, and transpiration changes therefore have a relatively lower impact.

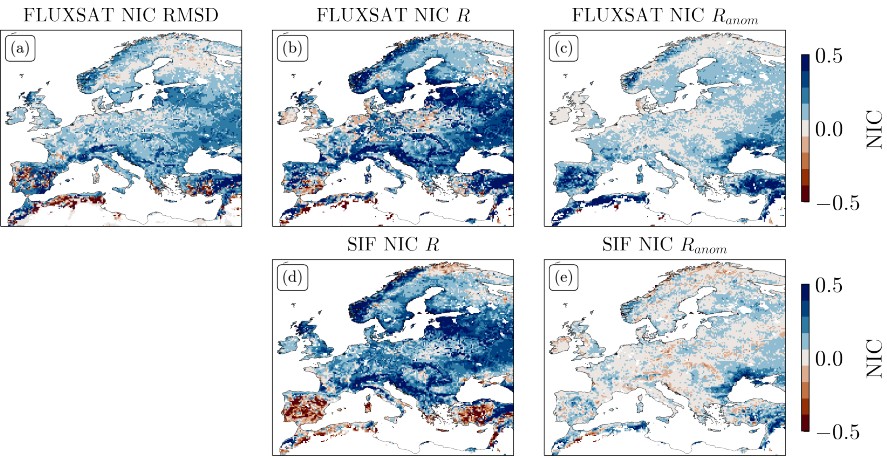

**Figure 2.** Maps of GPP NICs for the bias-blind DA for (a) RMSD with FluxSat, (b) $R$ with FluxSat, (c) $R_{anom}$ with FluxSat, (d) $R$ with SIF, and (e) $R_{anom}$ with SIF.

## 3.2 Evaluation of DA impacts on GPP

The impact of bias-blind and bias-aware LAI DA on GPP is shown in figures 2 and 3, respectively. Bias-blind LAI DA strongly improves GPP estimates in terms of RMSD and $R$ with FluxSat GPP and SIF (only $R$), over most of the domain, except in regions where the LAI bias is very large. In these regions, $R$ with SIF degrades almost everywhere, and GPP RMSD and $R$ with FluxSat degrades for some grid cells. The GPP $R_{anom}$ with FluxSat improves in most areas, especially in those with large LAI biases. Similarly, the highest improvements in $R_{anom}$ with SIF are found in areas with large LAI biases, excluding the Iberian Peninsula.

In the scaled LAI DA runs, the improvements in $R_{anom}$ are similar, but the improvements in RMSD and $R$ are lower, as summarised in Figure 3c and f. The CDF-matched DA improves GPP $R_{anom}$ with FluxSat over most regions, but not as strongly as the bias-blind DA (Figure 3a). The seasonally scaled DA has largest improvements in regions with large LAI bias, where it outperforms the CDF-matched DA, and has a low impact over the rest of the domain (Figure 3b). For SIF, the patterns in NIC $R_{anom}$ are similar for all three runs (Figure 3d-e).

## 3.3 Evaluation of DA impact hydrological variables

The impact of bias-blind and bias-aware LAI DA on hydrological ET and runoff fluxes is presented in Figures 4 and 5, respectively.

The ET shows mixed results in terms of RMSD, $R$, and $R_{anom}$ with GLEAM ET (Figure 4a-c). The bias-blind DA improves RMSD, $R$, and $R_{anom}$ over most of Turkey and the eastern Iberian Peninsula, but degrades over the western Iberian Peninsula and eastern Turkey. In central and eastern Europe, RMSD improves over most agricultural regions, but $R$ mostly degrades over these regions. In northern Europe, both RMSD and $R$ degrade compared to the OL run. The runoff estimates mainly improve

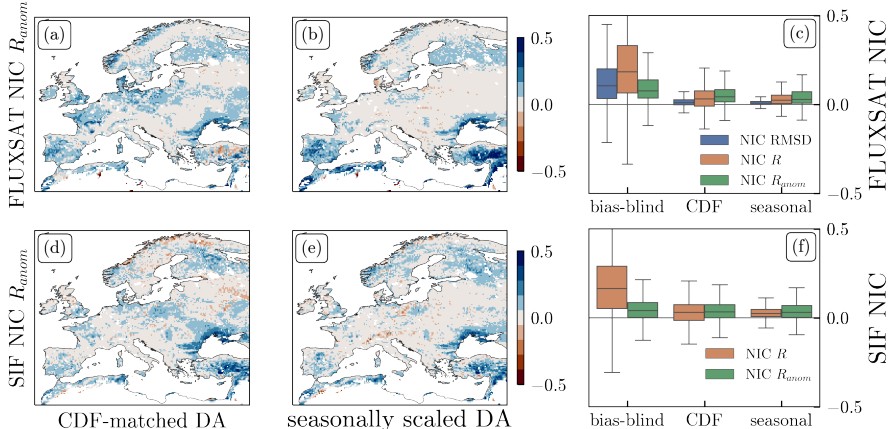

**Figure 3.** Top row: Maps of NIC $R_{anom}$ with FluxSat GPP for (a) the CDF-matched DA, (b) the seasonally scaled DA, and (c) box plots of NICs for RMSD, $R$, and $R_{anom}$ with FluxSat GPP for all three DA runs. Bottom row: Maps of NIC $R_{anom}$ with SIF for (d) the CDF-matched DA, (e) the seasonally scaled DA, and (f) box plots of NICs for $R$ and $R_{anom}$ with SIF for all three DA runs. The upper limit of the box plots showing NIC $R$ for the bias-blind DA (around 0.8 for FluxSat, 0.7 for SIF) has been cut here to facilitate a better comparison with the bias-aware runs.

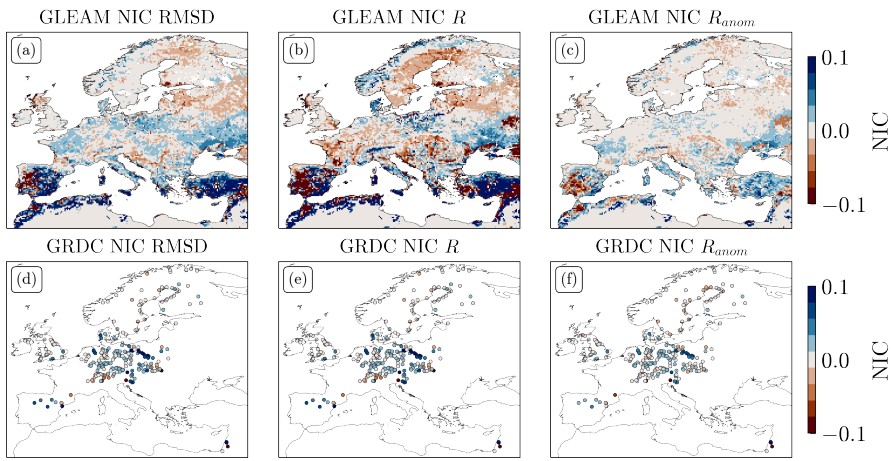

**Figure 4.** Top row: Maps of ET NICs for the bias-blind DA for (a) RMSD with GLEAM, (b) $R$ with GLEAM, and (c) $R_{anom}$ with GLEAM. Bottom row: Maps of runoff NICs for the bias-blind DA for (d) RMSD with GRDC, (e) $R$ with GRDC, and (f) $R_{anom}$ with GRDC. Note the different colour bar ranges compared to Figure 2

.

in terms of RMSD, $R$, and $R_{anom}$ with GRDC station data, especially in Spain and central Europe, but there is a negative

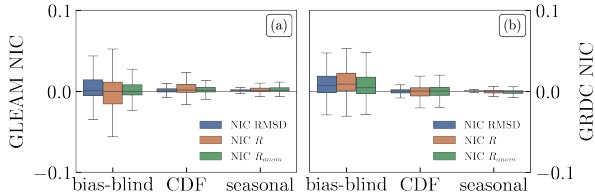

**Figure 5.** Box plots of RMSD, $R$, and $R_{anom}$ NICs for all three DA runs with (a) GLEAM ET and (b) GRDC runoff.

impact in the Alps and Scandinavia (Figure 4c-e). The rescaling techniques decrease both positive and negative DA impacts on ET and runoff, resulting in very low NICs (Figure 5a-b).

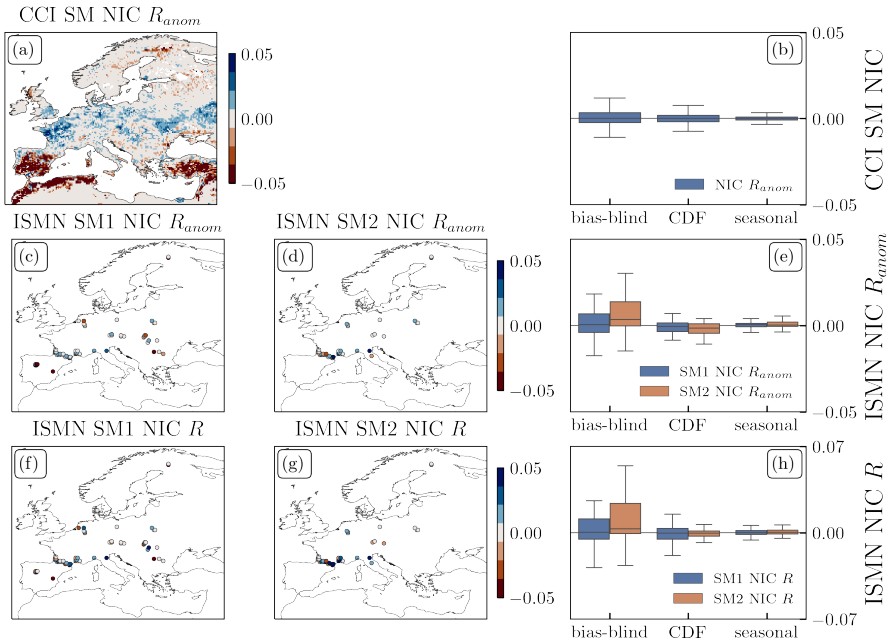

**Figure 6.** Top row: (a) Map of NIC $R_{anom}$ with ESA CCI SM for the bias-blind DA and (b) box plots of NIC $R_{anom}$ with ESA CCI SM for all three DA runs. Middle row: Maps of NIC $R_{anom}$ with ISMN for the bias-blind DA for (c) SM1 (0-10 cm) $R_{anom}$ and (d) SM2 (10-40 cm) $R_{anom}$, and (e) box plots of NIC $R_{anom}$ with ISMN for SM1 and SM2 and all three DA runs. Bottom row: Maps of NIC $R$ with ISMN for the the bias-blind DA for (f) SM1 (0-10 cm) $R$ and (g) SM2 (10-40 cm) $R$, and (h) box plots of NIC $R$ with ISMN for SM1 and SM2 and all three DA runs. Note the different colour bar range compared to Figure 2 and Figure 4.

380    Finally, the DA results are evaluated in terms of surface (0-10 cm) and deeper (10-40 cm) soil moisture against in situ data and the ESA CCI SM in Figure 6. The bias-blind DA leads to small improvements in deeper layer soil moisture in both $R$ and $R_{anom}$, but none of the included ISMN stations are in the areas with large biases. The bias-aware DA does not affect metrics with ISMN significantly.

The comparison with the satellite-based ESA CCI SM presents a spatially more complete picture, with $R_{anom}$ decrease in regions with large LAI bias (Figure 6d). $R_{anom}$ also decreases over several mountain ranges and in Scandinavia, but increases over agricultural areas in central Europe. The median NIC (Figure 6e) is small for all experiments, with a smaller NIC spread for the rescaled DA runs.

### 3.4   Example I: Majadas site

To interpret the strong relative differences found in the previous section, we confront time series of multiple model variables with in situ data for the Majadas site in Figure 7. We chose the years 2015 through 2017 as example, because of (1) the availability of in situ data, and (2) considerable interannual variability in OL and observed LAI.

The OL and CGLS LAI show some similar features in their temporal patterns, but the timing and magnitude disagree. Both show peaks in late spring or summer and reach their minimum in early autumn, followed by a small increase (Figure 7a). They also agree that the peak in spring/summer 2016 is the highest within these 3 years. However, the CGLS LAI reaches its maximum already start of May and then rapidly decreases, while the OL reaches its maximum later and decreases more slowly. Additionally, the OL has a higher overall LAI, and a lower interannual variation in maximum peak than the CGLS LAI. The magnitude of the spring maxima and the summer minima also match the observed maximum and minimum value better ($2.15\,\mathrm{m^2m^{-2}}$ and $0.55\,\mathrm{m^2m^{-2}}$, lower and upper thick grey line in Figure 7a, respectively). The large differences in summer lead to pronounced sawtooth patterns in the bias-blind DA results, showing that the model has a strong drift back towards the equilibrium state after each DA update.

The decrease of summer LAI in the DA also induces a decrease in summer GPP (Figure 7b). This increases $R$ with the in situ flux tower measurements, but slightly decreases $R_{anom}$. A better agreement can be seen in spring 2015, where observed and analysed GPP decline faster than the OL, and in spring 2017, where the OL GPP increases until mid May, while DA and observations stay at the same level as in April. The differences in overall magnitude between the in situ data and the model might be caused by representativeness errors, for example, differences in the assumed canopy cover for the savanna land cover class in the model and the canopy cover at the Majadas site.

Transpiration strongly decreases in summer as a consequence of the lower LAI (Figure 7c), which leads to a lower ET (Figure 7d). For the latter, correlation with the in situ data decreases, in agreement with the decreased correlation with GLEAM ET in the western Iberian Peninsula seen in Figure 4a, while the anomaly correlation slightly increases.

Soil moisture also increases, with a larger effect in the deeper layers (Figure 7e-g). The first layer (0-10 cm) is only slightly affected, but the deeper layers (layer 2 = 10-40 cm, layer 3 = 40-100 cm, layer 4 100-200 cm (not shown)) are much wetter in summer and autumn, caused by a slower drying rate. These large changes are hard to compare across scales, since the soil moisture climatology depends strongly on local factors like soil texture or topography (Dong and Ochsner, 2018).

The changes in the model LAI also affect surface and subsurface runoff (Figure 7h). The main difference in the example grid cell is an increased subsurface runoff for the analysis in winter 2016 and 2017.

Figure 8 shows that the two rescaling techniques studied in this paper reduce the difference between OL and analysis LAI. In the CDF-matched DA, winter LAI is higher than the OL, while in autumn LAI drops faster than in the OL. This leads to

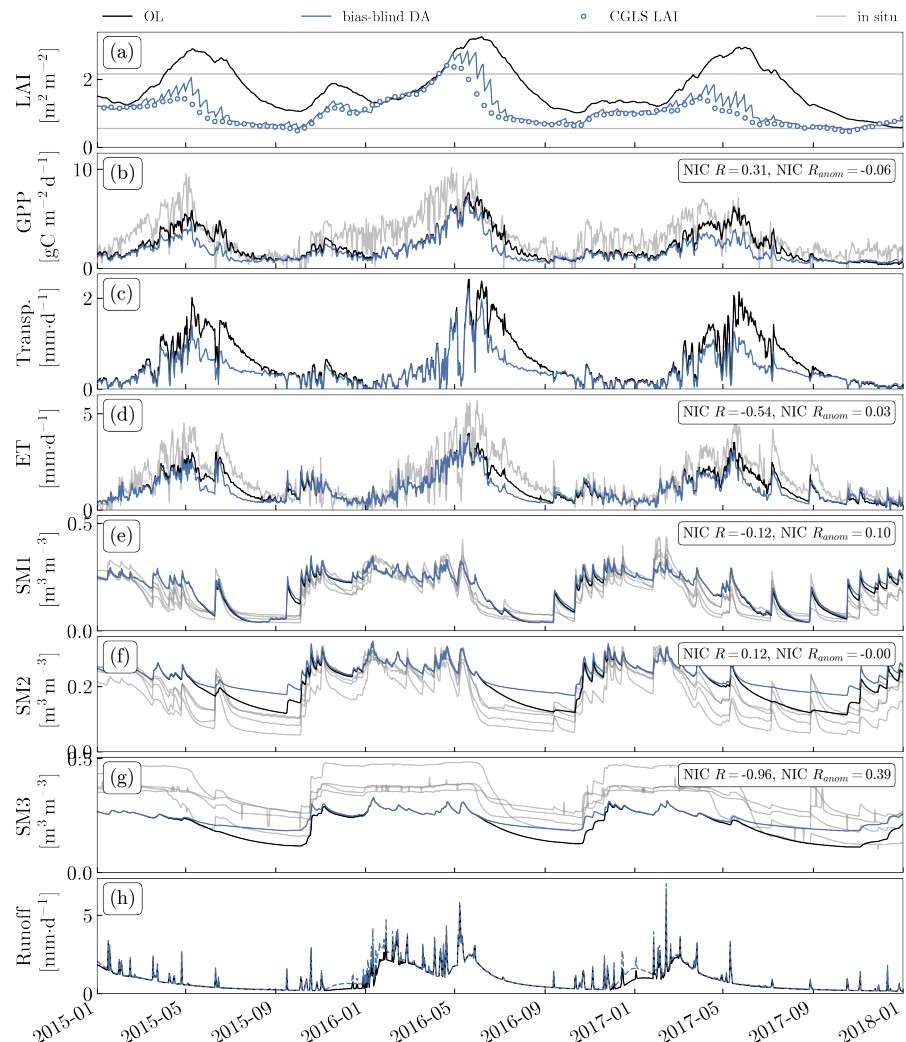

**Figure 7.** Time series of OL (black) and bias-blind DA (blue) results for (a) LAI, (b) GPP, (c) transpiration, (d) ET, (e) SM1 (0-10 cm), (f) SM2 (10-40 cm), (g) SM3 (40-100 cm), and (h) total runoff (surface + subsurface) for the model grid cell containing the Majadas site (39.875°, -5.875°). Panel (a) also shows the assimilated LAI observations (blue dots) and the minimum and maximum observed LAI at the site (grey lines). For the other panels, in situ data from the Majadas site are also shown (grey lines), if available, and the NICs for $R$ and $R_{anom}$ (calculated based on the full period of data availability) are indicated in the panels.

differences in layer 2 soil moisture in autumn, although they are not as strong as in the bias-blind DA. The seasonally scaled DA follows the OL more closely. The rescaled runs still contain the sawtooth pattern that was present in Figure 7a, but often with a less steep drift between updates, and with seasonally varying directions. Especially the seasonal rescaling performs well at reducing the sawtooth pattern.

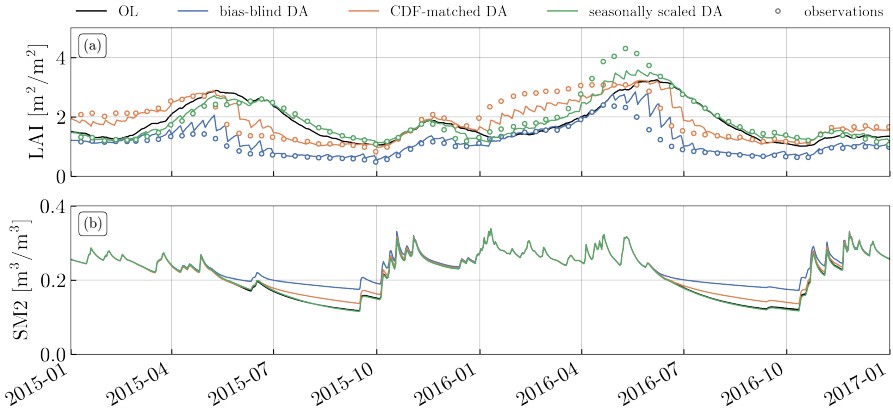

**Figure 8.** Time series of (a) LAI and (b) SM2 (10-40 cm) for all DA runs for the Majadas grid cell. Panel (a) includes the (potentially rescaled) observations that were assimilated in each run (coloured dots, dot colours correspond to line colours).

## 3.5 Example II: Nile delta

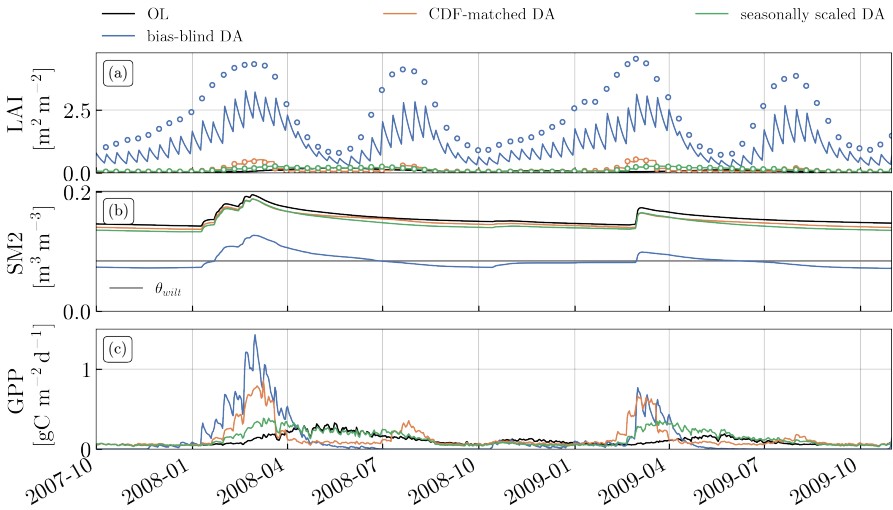

**Figure 9.** Time series of (a) LAI, (b) SM2 (10-40 cm), and (c) GPP for all DA runs for the Nile delta example grid cell ($31.125°, 30.875°$). Panel (a) includes the (potentially rescaled) observations that were assimilated in each run (coloured dots, dot colours correspond to line colours).

As another example we examined the Nile delta, where observed LAI strongly exceeds OL LAI, but summer GPP in the DA results strongly decreases compared to the OL (see Figure 1). The low vegetation in the OL is caused by a lack of irrigation in the model, which results in water limitations for vegetation growth. Figure 9a shows that the bias-blind DA strongly increases

LAI to follow the observations more closely. However, it also strongly decreases SM2 (Figure 9b), such that the wettest conditions in the bias-blind DA are still drier than the driest conditions in the OL. As a consequence, SM2 falls below the model wilting point in summer, and the model disables photosynthesis due to water stress (Figure 9c). This decouples analysed LAI and GPP in summer, and explains the decrease in April to October GPP seen in Figure 1. Instead of correcting the root cause for the LAI underestimation, the bias-blind DA worsens the problem here.

### 3.6   Analysis of equilibrium LAI

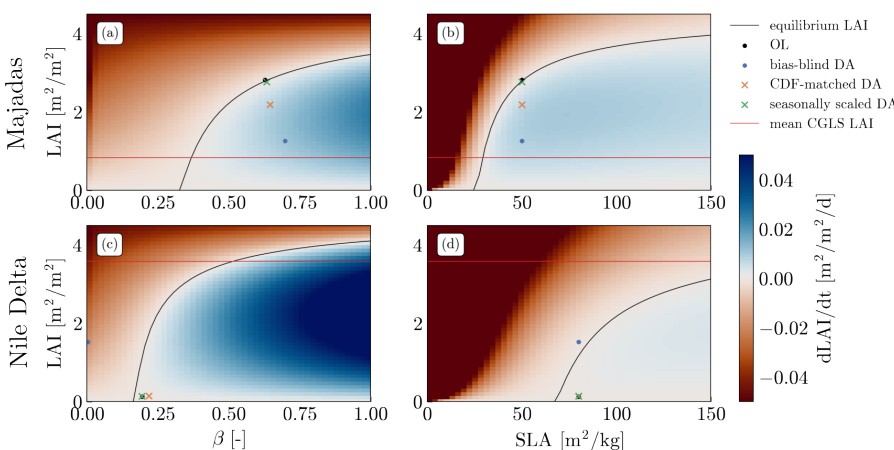

**Figure 10.** Estimates of the Noah-MP LAI change rate (dLAI/dt) and equilibrium LAI for the Majadas pixel in June (upper row), and the Nile delta pixel in July (lower row), as a function of the relative amount of plant available water $\beta$ (left column) and specific leaf area (SLA, right column). Additionally, the mean conditions for the OL and the DA runs are shown as symbols (dots, crosses), and the mean assimilated CGLS LAI as red line.

We assessed the dependency of the equilibrium LAI value on the model root-zone soil moisture (via the relative amount of plant available water $\beta$) and on model parameters (using the specific leaf area SLA as example parameter) for the two example sites discussed above.

Figure 10 shows how the equilibrium LAI would change if we would change $\beta$ or SLA while keeping everything else constant. For both shown sites (Majadas, Nile delta), we chose the month where the mean difference between OL and observations is largest (June for Majadas, July for Nile delta). We approximated the GPP-LAI relationship for these sites and months based on Equation B1.

For both considered cases, the mean OL conditions are close to the estimated equilibrium LAI, validating our approximations in the derivation of the method (section 2.6). For Majadas, the mean June conditions of the seasonally scaled DA are very close to the OL, while the bias-blind DA shows a strongly reduced LAI and an increased $\beta$, consistent with Figure 8. The CDF-matched DA is between OL and bias-blind DA. Both bias-blind and CDF-matched DA are further apart from the estimated equilibrium LAI than the seasonally scaled DA, i.e. they are not in a stable state. To obtain a stable state but at the same time

reduce the LAI towards the CGLS LAI observations, $\beta$ would have to be reduced to about 0.3, or alternatively SLA would have to be reduced to 30.

For the Nile delta pixel, both bias-aware DA runs are very close to the OL, while the bias-blind DA shows a strongly reduced $\beta$, consistent with summer conditions in Figure 9. In this pixel, the equilibrium LAI shows a much higher sensitivity on $\beta$; a small increase in $\beta$ would already lead to a large increase in LAI. Conversely, the sensitivity to SLA is low. To obtain a stable state close to the observed LAI, $\beta$ would have to be increased to 0.5, while SLA would have to be increased to values larger than 150.

## 3.7  Evaluation of short-term DA effects

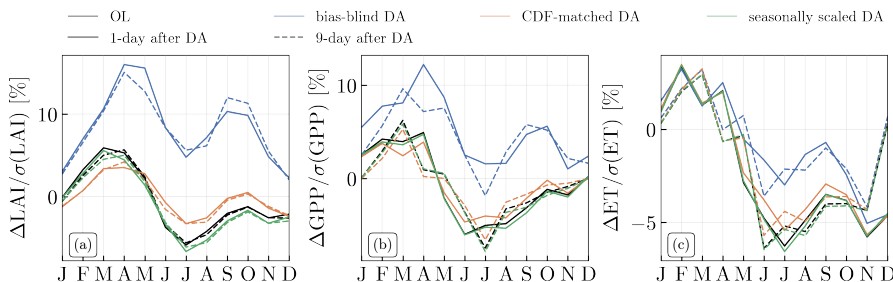

**Figure 11.** Normalised monthly median day-to-day forecast differences for (a) LAI, (b) GPP, and (c) ET. The differences are computed as the forecast value at 2-day after observations minus that of 1-day after observations ("1-day after DA", solid lines) and 1-day before observations minus 2-day before observations, corresponding to approximately 9-day after observations minus 8-day after observations ("9-day after DA", dashed lines) for the OL (black), the bias-blind DA (blue), the CDF-matched DA (orange) and the seasonally scaled DA (green). The median was calculated from all grid cells at which the relative LAI difference between OL and bias blind DA (see Figure 1a) is below -25%. For each grid cell and month, the median was normalised with the monthly standard deviation of the variable for this grid cell. The graph shows the median results across 17 years (2003-2019).

Figure 11 shows the monthly median day-to-day forecast differences for all performed simulation runs for LAI, GPP, and ET. The OL shows a seasonal cycle with high values in spring and low values in summer, as expected (corresponding to the derivative of the seasonal cycle of variable values). The bias-aware DA runs closely follow the OL seasonal cycle.

The bias-blind DA also shows the same seasonal cycle, but has an offset compared to the OL. For LAI, this offset is of the same size as the magnitude of the mean seasonal cycle, so that the mean day-to-day differences in the bias-blind DA in summer have the same magnitude as the day-to-day differences in the OL in spring, even though physically a decrease in LAI is expected. In fact, the day-to-day differences in LAI in the bias-blind DA are always positive, meaning that LAI is expected to increase in all seasons. This is caused by the large DA update steps in the bias-blind DA, which pull the model to an unstable state. As a consequence, model instability instead of physical forcing input governs the short-term temporal evolution of LAI in the model in between update steps. Even 9 days after the DA update, right before the next update step, day-to-day differences do not significantly change, indicating that the model instability can persist for long time periods.

The instability effect also strongly affects GPP estimates throughout all seasons, and to a lesser extent ET estimates in summer.

## 3.8 DA diagnostics

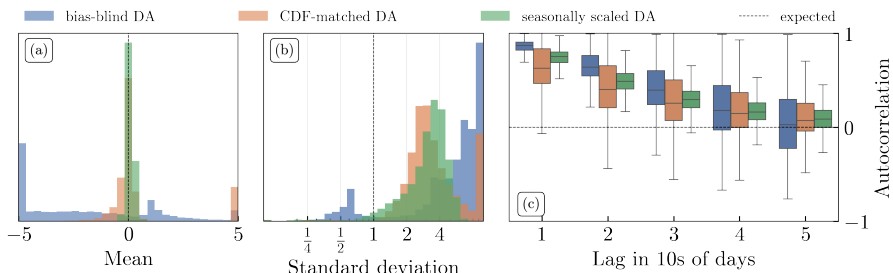

**Figure 12.** Spatial distributions of the temporal (a) mean, (b) standard deviation, and (c) autocorrelation of the innovations, across all model grid cells. (a-b) Values for means and standard deviations outside the plot range of the histograms have been added to the first and last bin, respectively. (c) The autocorrelation is computed for multiple lags of 10 days.

Figure 12 shows distributions of innovation statistics across the modelling domain and shows that the innovation sequence is not standard normal for the bias-blind DA. As a consequence of the higher LAI in the model, the normalised innovation mean is strongly negative (Figure 12a), and the absolute values of the innovations are large (Figure 12b). The autocorrelation is also high (Figure 12c) because subsequent updates point in the same direction.

The rescaling improves the internal diagnostics of the DA system. Although there is still a sawtooth pattern (Figure 8), the assumption of zero mean innovations is met and rescaling helps to reduce the innovation variance (Figure 12b) and the autocorrelation (Figure 12c) compared to the bias-blind DA run.

## 4 Discussion

### 4.1 General impacts of bias-blind and bias-aware DA

Our analysis shows that large biases between Noah-MP modelled LAI and CGLS LAI exist. This includes bias in the length of the growing season, which might be caused by processes not included in the model (e.g. agriculture) or biased forcing data, but also strong bias in the CGLS LAI magnitude. It is most pronounced over dry areas in the southern part of the modelling domain, in line with results of Li et al. (2022), who also found an overestimation of LAI by Noah-MP's dynamic vegetation model with respect to MODIS LAI in this area. Noah-MP is not unique in this respect; studies with other LSMs have also found model deficiencies in dry regions (Dahlin et al., 2015; MacBean et al., 2015; Fox et al., 2018; Mahmud et al., 2021).

The bias-blind LAI DA therefore has a strong impact on the vegetation model state and fluxes. Where LAI bias is large, the bias-blind DA induces strong changes in GPP magnitude, which are mostly reducing RMSD with FluxSat, in agreement

with results found by Kumar et al. (2019b) and Albergel et al. (2020) for similar GPP reference datasets. Anomaly correlation improvement for FluxSat and SIF differs, but both show generally a positive impact. The difference might be due to the dependence of both the assimilated LAI observations and the FluxSat GPP retrievals on reflectances from optical satellite sensors, which might inflate anomaly correlations.

The strong impacts of the bias-blind DA also propagate to the model hydrology. Results for ET estimates are mixed: RMSD and $R_{anom}$ with GLEAM generally improve, especially over Turkey, the western Iberian Peninsula, and agricultural regions, but $R$ deteriorates over most of the domain. In contrast, runoff estimates improve compared to the GRDC discharge data. The comparison to ISMN indicates improvements in deeper layer soil moisture, but none of the in situ sites considered are in the areas with large bias. Anomaly correlation with ESA CCI SM also improves over agricultural regions, but decreases over high-bias regions and northeastern Europe. However, in northeastern Europe the Noah-MP model-only SM estimates outperform ESA CCI SM when comparing to in situ sites (Heyvaert et al., 2023), probably due to the lower signal-to-noise ratio of soil moisture retrievals over dense vegetation (Gruber et al., 2019).

The large changes to the root-zone soil moisture climatology are hard to assess directly, because of the scale difference between in-situ data and model grid cells, and the lack of in situ sites in these areas. However, in strongly irrigated areas the change in soil moisture climatology leads to a decrease in soil moisture, even though the bad model performance originates from an underestimation of soil moisture due to the lack of an irrigation process in the model. Joint updates of LAI and root-zone soil moisture as done in LDAS-Monde (Albergel et al., 2017) could alleviate this problem caused by "missing" water to some extent but require a good estimation of the coupling strength of LAI and soil moisture. The strong effect on the model hydrology might also be model-specific, because the Noah-MP model hydrology is more sensitive to vegetation than other LSMs (Maertens et al., 2021).

Even though our results for RMSD improvements in GPP and ET are similar to other studies (Kumar et al., 2019b; Albergel et al., 2020), it is important to note that none of the reference products we used are free of bias. This can be due to assumptions and errors in the underlying satellite data and retrieval algorithms in the case of satellite-based data, or due to different spatial support in the case of in situ data. Hence, whether the bias-blind DA leads to estimates closer to the "truth" remains uncertain, and evaluations with different reference products might come to different conclusions. We therefore additionally investigated effects of the DA on the model and on internal DA diagnostics.

## 4.2 Negative effects on the optimality of the DA system when ignoring bias

As an effect of "misusing" a Kalman filter for correcting biases instead of random errors, DA updates are strongly biased, leading to non-optimal DA diagnostics and a pronounced sawtooth pattern. Such sawtooth patterns are common in filter DA (e.g., Mitchell et al., 2002; Dee, 2005; Fox et al., 2018), but the strong preference for one direction and the model drift between two update steps harms estimates of other variables. GPP and ET are strongly reduced at the time of the DA update, and the short term model forecasts directly after the DA update step show unphysical upward drifts.

The sawtooth pattern also poses the danger of introducing spurious trends in case the observation frequency changes over time (Dee, 2005). For example, if the availability of LAI observations at the Majadas site increased over time, the model would

be pulled more closely to the observations, i.e. lower LAI values in the later periods than in the early period, leading to an apparent decrease in LAI. Due to the strong impact of LAI changes on soil moisture, this would also lead to a spurious wetting trend in the deeper soil layers. Such artificial trends can seriously confound trends in the resulting dataset.

The sawtooth pattern has also been reported by other LAI DA studies using Noah-MP (Kumar et al., 2019b; Mocko et al., 2021). We showed that the Noah-MP leaf model structure has an equilibrium LAI independent of the current model state, to which the model tries to return after each update step. Bias-blind LAI DA into LSMs with a similar model structure might therefore suffer from the similar issues.

    The sawtooth pattern can be reduced by interpolating the observations or by applying time series smoothing methods to
obtain pseudo-observations at a daily frequency. This will keep the analysis closer to the observations and prevent model drift over multiple days. However, in this case, direct insertion approaches or using observed LAI directly as a model parameter could achieve even better results than an EnKF, at a much lower computational cost.

## 4.3   Effects of bias-aware DA

Using rescaling techniques for a priori bias correction comes at the cost of foregoing improvements in ET, runoff, and GPP.
However, the rescaling techniques retain improvements in GPP anomaly correlation and limit the side effects of the LAI DA on model hydrology. The CDF-matching performs better for GPP anomalies over central Europe and for ET over the high-bias regions, since it preserves more information on the shape of the observed seasonal cycle of LAI. But due to its larger impact on ET compared to the seasonal rescaling it also changes the soil moisture climatology in deeper layers, and leads to a decrease in anomaly correlation with in situ soil moisture, especially for deeper layers. The seasonal rescaling has a better performance
for GPP anomalies over the high-bias regions when using FluxSat GPP as reference, but not with SIF, which may indicate an overestimation in skill as discussed above. Overall, the seasonal rescaling minimises the DA effects on the model hydrology.

    Since the bias-aware DA limits the DA to address only the random error components, filter diagnostics are more in line with standard assumptions (Desroziers et al., 2005). This facilitates a further reduction of the variance and autocorrelation of the normalised innovations to obtain an optimal filter configuration by tuning the model and observation perturbations.

The limited DA impacts and more well-behaved filter performance could be especially helpful when assimilating multiple datasets, because contrasting biases could deteriorate the ability of the DA system to find a good compromise between multiple observations and model predictions (MacBean et al., 2016). This might for example arise if variables that require more complex observation operators are assimilated, and the observation operator is calibrated to the original model climatology.

## 4.4   Alternatives to rescaling of observations

The bias-aware DA uses observation rescaling methods to reduce the effects of biased updates like drifts and sawtooth patterns, but leads to lower improvements than the bias-blind DA, because it leads to LAI estimates in the model climatology. To obtain LAI estimates within the range of the observational LAI climatology while still keeping the model in a stable state, additional model state variables or parameters have to be updated or calibrated, or an observation or forecast bias could be estimated separately (and removed from the innovations) (De Lannoy et al., 2007a, b).

Calibration of model parameters is the best option if the bias is due to uncertainty in the model parameters. It has been successful at improving vegetation models in previous studies MacBean et al. (2015, 2016); Scholze et al. (2019); Forkel et al. (2019); Kolassa et al. (2020); Mahmud et al. (2021). Updates of specific leaf area together with LAI have already been used successfully by Xu et al. (2021) and He et al. (2022). Based on Equation A1, changes in the leaf turnover coefficient or the respiration coefficient might lead to similar results.

The calibration can either be done prior to the DA simulations or can be incorporated into a joint parameter and state update DA scheme. An EnKF (as used in this study) can in principle be used for the joint updates by augmenting the control vector to contain both state variables and parameters (Evensen, 2009). If the model predictions' dependency on the parameters is highly nonlinear, particle methods might be more suitable (Frei and Künsch, 2013; van Leeuwen et al., 2019). Hybrid methods that combine the EnKF with particle methods could be used to obtain a DA system that performs well both for state updates and 560 parameter updates (Frei and Künsch, 2013; van Leeuwen et al., 2019; De Lannoy et al., 2022).

Instead, if the (short-term) bias is caused by erroneous forcing data, e.g. a seasonal wet or dry bias in water-limited areas, joint updates of LAI and RZSM are better suited to improve the analysis. This is the approach chosen for (bias-blind) LAI DA in LDAS-Monde (Albergel et al., 2017). With an EnKF, joint updates are achieved via error cross-correlations between LAI and RZSM (obtained from the ensemble). This means that RZSM will only be updated in phases in which model LAI and 565 RZSM show a strong coupling.

A known source of bias in the Nile delta is the missing irrigation input in the model. Additionally, we found a very strong water limitation on vegetation growth, and a strong sensitivity of the equilibrium LAI on changes in RZSM, implying a strong model coupling of LAI and RZSM. Therefore, joint updates of LAI and RZSM are likely to improve the LAI DA results here because RZSM is temporarily adjusted, but changes to soil parameters might also be necessary to sustain the increased 570 moisture values.

At the Majadas site, the sensitivity of the equilibrium LAI to changes in RZSM is lower. Furthermore, Figure 8 indicates that in winter and spring, RZSM is largely dominated by soil parameters and precipitation input. In these periods, changes in LAI do not lead to changes in RZSM, and differences in RZSM between the OL and the bias-blind DA vanish quickly. Therefore, RZSM updates are unlikely to sustainably decrease RZSM during spring, which would be required to decrease LAI to a more 575 stable state. This could make joint updates of LAI and RZSM less efficient than in the Nile delta. Parameter updates are likely most useful for improving the LAI estimates at the Majadas site.

## 4.5 Potential model structural changes

The vegetation model in Noah-MP consists of two parts: a photosynthesis model, which calculates how much carbon is assimilated from the atmosphere in each time step, and the dynamic vegetation model, which distributes the carbon to different plant 580 carbon pools and calculates losses due to respiration and turnover. Previous studies found that the dynamic leaf model decreases performance compared to a prescribed LAI (Ma et al., 2017; Erlingis et al., 2021; Huang et al., 2022). Structural changes in the equations governing the leaf carbon assimilation might therefore improve the agreement of modelled and observed LAI.

A promising candidate for structural changes is the leaf carbon allocation function, which governs which fraction of the photosynthesis carbon is allocated to the leaves. In Noah-MP v4.0.1, this function decreases from 1 at LAI=0 to 0 at approximately LAI=6 with a sigmoidal-like shape. Alternative formulations have been tested by Gim et al. (2017) and Niu et al. (2020). They used sigmoidal functions with a sharp decline around a threshold LAI. This would increase the model drift towards the equilibrium (the threshold LAI), and therefore likely worsen the instability in a bias-blind DA setup. But when treating this threshold as a model parameter, these formulations open up new possibilities for calibration and parameter data assimilation, since the threshold LAI gives a more direct access to adapting the maximum LAI reached in summer. Multi-pass schemes that update the threshold based on observations, similar to Xu et al. (2021), might be able to improve the persistence of observations and alleviate the sawtooth pattern issue.

Another shortcoming of Noah-MP is its oversimplified phenology scheme, which is solely based on a land cover-specific canopy temperature threshold, ignoring other drivers of phenology like day length or water availability (e.g., Dahlin et al., 2015, 2017), or cumulative temperature effects often expressed via growing degree days (e.g., in CLM, Lawrence et al., 2011). Especially in the southern part of our modelling domain, where water partly limits vegetation growth (Hashimoto et al., 2019), more complex phenology schemes might improve the realism of the vegetation simulations. In the current scheme, the temperature threshold is almost always exceeded, leading to unrealistically long growing seasons. However, additional degrees of freedom introduced by a more complex phenology scheme can also deteriorate model predictions (Lawrence et al., 2011).

## 5 Conclusions

So far, satellite LAI DA studies have mostly ignored biases between observed and modelled LAI. In this study, we evaluated how the presence of bias in an LAI DA system can impact the model hydrology and carbon uptake. Specifically, we assimilated CGLS LAI into Noah-MP with an EnKF, and we evaluated a bias-blind DA and two rescaling techniques, i.e. climatological CDF-matching and seasonal rescaling of the first two moments, to account for the biases in the DA system.

The bias-blind DA is most effective at reducing the disagreement in modelled and observed LAI, and leads to the largest improvements in GPP and runoff. It is therefore a suitable option for many applications, especially if large bias reductions are intended, even though bias-blind Kalman filtering is suboptimal. A temporal interpolation of the observation data, or even a direct insertion approach, could be even more efficient for such a purpose. However, this approach does not necessarily improve other variables, e.g. if the model simulates biased LAI in conjunction with unbiased soil moisture. As an alternative, we recommend using observation rescaling techniques for LAI DA with Noah-MP if there are strong biases and if

– the focus is not only on vegetation or the carbon cycle, but also on hydrological processes, because large LAI changes can cause unphysical impacts on model hydrology;

– multiple datasets with contrasting biases are assimilated, since the bias-blind DA can strongly change the model climatology;

- the DA aims at preparing the best analysis state for subsequent short-term predictions, because the abrupt update steps induce spurious short-term trends;

- datasets with changes in observation frequency are used, because this can induce spurious long-term trends;

- an optimal DA system in terms of Desrozier's metrics (Desroziers et al., 2005) is desired, because bias-blind DA violates basic assumptions of the Kalman filter.

The CDF-matching technique preserves more information from the signal and leads to larger improvements in GPP and ET, but worse estimates of deeper layer soil moisture. The seasonal rescaling is more effective at removing bias and limits DA updates to improve vegetation anomalies; it performs best in terms of internal DA diagnostics. The bias-aware LAI DA is suitable for providing physically consistent short-term flux estimates for numerical weather prediction models or soil moisture monitoring, or a baseline to merge historical earth observation records from multiple sensors to a long-term dataset without introducing artificial trends.

A drawback of the observation rescaling approaches is that they result in estimates in the model climatology. If the observation-forecast bias is due to erroneous precipitation forcing or missing irrigation input, joint updates of LAI and RZSM in a bias-blind system can be considered instead. This might lead to large bias corrections while still retaining a stable model state even after large updates. However, if the bias is not only caused by bias in the precipitation/irrigation, this poses the risk of seriously degrading the soil moisture estimates.

Alternatively, updates to model parameters, either via joint parameter and state update DA, or via a priori model calibration, can also lead to more stable and persistent updates and LAI estimates in the observational climatology. This is especially desirable for research on the carbon cycle, where absolute values of carbon fluxes are required. Parameters to consider for calibration are parameters related to model leaf growth, but potentially also photosynthesis or soil parameters.

To gain the most benefit from LAI data assimilation into Noah-MP, further research and model improvement of the coupling mechanisms between the water and carbon cycle is necessary.

## Appendix A: Noah-MP dynamic leaf model

This section gives a short overview of the Noah-MP vegetation model, focusing on the interaction of LAI and soil moisture. For a more detailed description we refer to Niu et al. (2011).

Noah-MP calculates LAI from a prognostic leaf biomass $C_l$ and a vegetation-type specific leaf area per leaf mass (specific leaf area; SLA):

$$LAI = \text{SLA} \cdot C_l$$

Leaf biomass is updated in each step via a mass balance equation:

$$\frac{\mathrm{d}C_l}{\mathrm{d}t} = \frac{1}{\mathrm{SLA}} \frac{\mathrm{d}LAI}{\mathrm{d}t} = (1 - \mathrm{FRAGR}) \cdot [f_l(LAI) \cdot GPP(LAI, \beta, T_c, F) - R_m(LAI, \beta, T_c)]$$
$$- D_c(LAI, T_c) - D_d(LAI, \beta) - T_l(LAI) \tag{A1}$$

with FRAGR the fraction of GPP minus maintenance respiration invested in growth respiration, $R_m$ the maintenance respiration, $D_c$ the death rate due to cold stress, $D_d$ the death rate due to drought stress, and $T_l$ the turnover rate, $F$ the atmospheric forcings, and $T_c$ the canopy temperature. $f_l$ is the fraction leaf carbon allocation, which governs how much of the total assimilated carbon (GPP) is allocated to the leaf pool (the rest will be allocated to other carbon pools) and is given by

$$f_l(LAI) = \left(1 - \frac{LAI}{10}\right) \cdot \exp\left(0.01 \cdot LAI \left(1 - e^{0.75 \cdot LAI}\right)\right)$$

The dependence of vegetation growth on available soil moisture is controlled via the soil moisture factor $\beta$, which represents the relative amount of plant available water in the root zone:

$$\beta = \sum_i \frac{\Delta z_i}{z_{root}} \min\left(1, \frac{\theta_i - \theta_{wilt}}{\theta_{fc} - \theta_{wilt}}\right)$$

where $\theta_i$ is the volumetric soil moisture in layer $i$, $\theta_{fc}$ is the field capacity, and $\theta_{wilt}$ is the wilting point.

The sink terms of Equation A1 are calculated in the following way:

$$R_m(LAI, \beta, T_c) = 12 \times 10^{-6} \cdot R_{25} \cdot \mathrm{FNF} \cdot 2^{\frac{T_c - 298.16}{10}} \cdot \beta \cdot LAI \tag{A2}$$

$$D_c(LAI, T_c) = 10^{-6} \cdot \frac{LAI^2}{120 \cdot \mathrm{SLA}^2} \cdot c_c \exp\left(-0.3 \max(0, T_c - T_{c,min})\right) \tag{A3}$$

$$D_d(LAI, \beta) = 10^{-6} \cdot \frac{LAI}{\mathrm{SLA}} \cdot c_d \cdot \exp\left(-100\beta\right) \tag{A4}$$

$$T_l(LAI) = 5 \times 10^{-7} \cdot c_t \cdot \frac{LAI}{\mathrm{SLA}} \tag{A5}$$

with land cover specific parameters $R_{25}$ (maintenance respiration at 25°C), FNF (foliage nitrogen factor), $c_c$ (cold stress coefficient), $T_{c,min}$ (leaf freezing temperature), $c_d$ (drought stress coefficient), $c_t$ (turnover coefficent).

## Appendix B: Climatological approximation of the Noah-MP LAI equilibrium value as function of soil moisture and leaf parameters

The equilibrium LAI value (model-based 'attractor') is the LAI value at which Equation A1 is zero. It is therefore an implicit function of all the terms and variables on the right hand side of Equation A1. However, some of the terms on the right hand side of this equation strongly depend on the meteorological forcings (e.g., GPP). Evaluating the equilibrium LAI as function of

soil moisture and leaf parameters would therefore require running the complete Noah-MP model with a wide range of forcing conditions, which quickly becomes computationally intractable. Therefore, we use a climatological approximation of the term in Equation A1 to eliminate the explicit dependence on the meteorological forcings.

We obtain a climatological approximation of GPP as a function of LAI and $\beta$ by assuming that GPP is proportional to LAI, $\beta$, and a factor that depends solely on the forcings $F$ or constant parameters not including the leaf parameters:

$$GPP(LAI, \beta) \approx LAI \cdot \beta \cdot \alpha(F) \tag{B1}$$

The linear dependence on $\beta$ is part of the Noah-MP model physics, while the assumption of linear dependence on LAI is justified in case vegetation growth is not light limited. This is reasonable for the areas with a large bias in the southern part of the domain, where vegetation growth is mainly water limited. To find an approximation for $\alpha$, we perform a least-squares fit of Equation B1 using daily mean model output for GPP, LAI, and $\beta$ for each calendar month. This results in 12 separate approximations of $GPP(LAI, \beta)$, one for each calendar month. The fit for the month with the highest discrepancy between Noah-MP OL and CGLS are shown in Figure B1

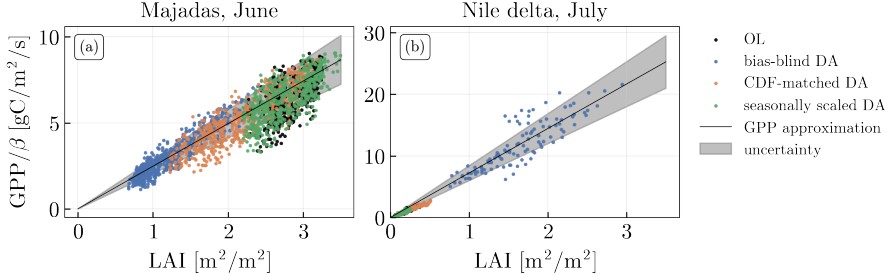

**Figure B1.** Dependence of soil-moisture-normalized GPP (GPP/$\beta$) on LAI for the OL and DA runs, and linear approximation via least-squares fit for (a) Majadas in June and (b) the Nile delta in July.

For the other terms in Equation A1, we simply insert the mean forcing value if required. The resulting defining equation for the equilibrium LAI for the month $m$ is then

$$\begin{aligned}
0 = & \beta_m \cdot (1 - \text{FRAGR}) \cdot [f_l(LAI_{eq,m}) \cdot \alpha_m \cdot LAI_{eq,m} - R_{m,w}(LAI_{eq,m}, T_{c,m})] \\
& - D_c(LAI_{eq,m}, T_{c,m}) - D_d(LAI_{eq,m}, \beta) - T_l(LAI_{eq,m})
\end{aligned} \tag{B2}$$

where $T_{c,m}$ is the mean canopy temperature, $\beta_m$ the mean plant available water, and $\alpha_m$ the GPP proportionality factor from Equation B1 for month $m$, respectively. The solution can be obtained numerically with common root-finding algorithms.

**Table C1.** ISMN networks used for evaluation.

| Network name | Country | Stations | Coverage | References/Acknowledgements |
|---|---|---|---|---|
| CALABRIA | Italy | 5 | 2001-2012 | Brocca et al. (2011b) |
| CAMPANIA | Italy | 2 | 2000-2012 | Brocca et al. (2011b) |
| COSMOS | Switzerland | 1 | 2008-2020 | Zreda et al. (2008, 2012) |
| FMI | Finland | 27 | 2007-2022 | Ikonen et al. (2016, 2018) |
| FR_Aqui | France | 5 | 2012 - 2022 | Al-Yaari et al. (2018); Wigneron et al. (2018) |
| GTK | Finland | 7 | 2001 - 2012 | Raimo Sutinen |
| HOAL | Austria | 33 | 2013-2021 | Blöschl et al. (2016); Vreugdenhil et al. (2013) |
| HOBE | Denmark | 32 | 2009-2019 | Jensen and Refsgaard (2018); Bircher et al. (2012) |
| HYDROL-NET_PERUGIA | Italy | 2 | 2010 - 2016 | Morbidelli et al. (2017) |
| IMA_CAN1 | Italy | 12 | 2011-2015 | Biddoccu et al. (2016); Raffelli et al. (2017) |
| IPE | Spain | 2 | 2008-2020 | Alday et al. (2020) |
| MOL-RAO | Germany | 2 | 2003-2020 | Beyrich and Adam (2007) |
| NVE | Norway | 3 | 2012 - 2019 | Norwegian water resources and energy directorate (NVE), Fred Wenger |
| ORACLE | France | 6 | 1985 - 2013 | Institut national de recherce en sciences et technologies pour l'environment et l'agriculture France |
| REMEDHUS | Spain | 24 | 2005-2022 | González-Zamora et al. (2019) |
| RSMN | Romania | 20 | 2014 - 2022 | Romanian National Meteorological Administration, Andrei Dimandi, Adelina Mihai |
| SMOSMANIA | France | 22 | 2007 - 2021 | Calvet et al. (2016); Albergel et al. (2008); Calvet et al. (2007) |
| STEMS | Italy | 4 | 2015 - 2022 | Capello et al. (2019); Darouich et al. (2022) |
| SWEX_POLAND | Poland | 6 | 2000 - 2013 | Marczewski et al. (2010) |
| TERENO | Germany | 5 | 2009-2021 | Zacharias et al. (2011); Bogena et al. (2018, 2012); Bogena (2016) |
| UDC_SMOS | Germany | 11 | 2007 - 2011 | Schlenz et al. (2012); Loew et al. (2009) |
| UMBRIA | Italy | 13 | 2002 - 2017 | Brocca et al. (2011a, 2008, 2009) |
| UMSUOL | Italy | 1 | 2009 - 2017 | Agenzia Regionale Prevenzione Ambiente - Servizio Idro-Meteo-Clima (ARPA - SIMC) and Andrea Pasquali |
| WEGENERNET | Austria | 12 | 2007 - 2022 | Fuchsberger et al. (2021); Kirchengast et al. (2014) |
| WSMN | UK | 8 | 2011 - 2016 | Petropoulos and McCalmont (2017) |

# Appendix C: Used ISMN networks

685 *Author contributions.* Samuel Scherrer performed the data assimilation runs and the analysis of the results, and drafted the manuscript. Zdenko Heyvaert, Gabriëlle De Lannoy, and Michel Bechtold assisted with the setup of the LSM and the data assimilation system. Zdenko Heyvaert, Gabriëlle De Lannoy, and Michel Bechtold, Wouter Dorigo and Clement Albergel provided scientific input to the design of the study. Tarek S. El-Madany provided in situ data. All authors contributed to the final draft of the paper by providing input for the final manuscript and discussion of the results.

690 *Competing interests.* The authors declare that they have no conflict of interest.

*Disclaimer.*

*Acknowledgements.* This work was funded by FWF (Austrian Science Fund) and FWO (Research Foundation Flanders) as part of the CON-SOLIDATION project (G0A7320N). We also acknowledge support from the European Space Agency Climate Change Initiative (Contract No. 4000126684/19/I-655 CCI+ Phase 2) and Land Carbon Constellation (Contract No. 4000131497/20/NL/CT) projects. Computing re-
695 sources were provided by the Flemish Super Computer Center (VSC). The authors acknowledge TU Wien Bibliothek for financial support through its Open Access Funding Programme. We thank two anonymous reviewers for their reviews, which helped to greatly improve this manuscript.

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
