# Peer review of "Bias-blind and bias-aware assimilation of leaf area index into the Noah-MP land surface model over Europe"

_EGUsphere, 2022_

## Author Comment (AC1)

**Reply to Comment by Anonymous Referee #1**

April 3, 2023

We thank the anonymous reviewer for the very thorough and well put comment. We will start by addressing the general comments for the larger issues, and afterwards address the particular comments.

In the following, review comments will be styled in *blue italics*, our responses in black normal text, and proposed changes in green.

**1  General comments**

**1.1  "Logic issue"**

*Logic: the Authors recommend dropping their most efficient bias-blind LAI DA option using specious theoretical arguments.*

The main novelty of our study is the impact of bias-aware DA methods for LAI DA and an assessment under which circumstances this might be helpful. We therefore focused mainly on pointing these out in the conclusions.

However, based on this and the other review comment we see that this can be easily misunderstood. We therefore propose to include the following sentence in the conclusion after l.549:

The bias-blind DA is most effective at reducing the disagreement in modelled and observed LAI, and leads to the largest improvements in GPP and runoff. For many applications it is therefore a suitable option, especially if large bias reductions are intended, even if bias-blind Kalman filtering is suboptimal. A temporal interpolation of the observation data, or even a direct insertion approach, could be even more efficient in this case. However, this approach does not necessarily improve other variables, e.g. if the model simulates biased LAI in conjunction with unbiased soil moisture. As alternative, we recommend to use observation rescaling techniques for LAI DA with Noah-MP if there are strong biases and if

- ... (same as in original manuscript)

Additionally, we will reformulate the last paragraph of the conclusion as follows:

A drawback of the observation rescaling approaches is that they result in estimates in the model climatology. If the observation-forecast bias is due to erroneous precipitation forcing or missing irrigation input, joint updates of LAI and RZSM in a bias-blind system can be considered instead. This might lead to large bias corrections while still retaining a stable model state even after large

updates. However, if the bias is not only caused by bias in the precipitation/irrigation, this poses the risk of seriously degrading the soil moisture estimates.

Alternatively, updates to model parameters, either via joint parameter and state update DA, or via a priori model calibration can also lead to more stable and persistent updates and LAI estimates in the observational climatology. This is especially desirable for research on the carbon cycle, where absolute values of carbon fluxes are required. Parameters to consider for calibration are parameters related to model leaf growth, but potentially also photosynthesis or soil parameters.

To gain the most benefit from LAI data assimilation into Noah-MP, further research and model improvement of the coupling mechanisms between water and carbon cycle is necessary.

**1.2 "Scope issue"**

*Scope: both Abstract and Conclusion sections present results that are specific to the Noah-MP model and DA framework as if they were valid for all models and DA tools. Much different conclusions could have been obtained using other tools, and probably more logical conclusions. The title is also too general.*

We agree that different conclusions might have been obtained with different models or different DA systems. We mentioned this in the discussion section (l. 462-465 or the more detailed discussion of shortcomings of the Noah-MP vegetation model in subsection 4.4 of the manuscript). In a revised version, we will provide a more detailed discussion of the Noah-MP model (see below). This will make it easier to transfer our results to other models or datasets, because it makes the underlying reasons in the model more transparent.

Additionally, we will adapt the title and the abstract as follows:

New title: Bias-blind and bias-aware LAI data assimilation over Europe with Noah-MP

In the abstract, we will modify l.12-15 as follows:

Furthermore, the bias-blind LAI DA produces a pronounced sawtooth pattern due to model drift between update steps, because the update step pushes the Noah-MP leaf model to an unstable state. This model drift also propagates to short-term estimates of GPP and ET, and to internal DA diagnostics that indicate a suboptimal DA system performance.

**1.3 Validation data quality**

*Basic hypotheses on the validation data used to perform the analysis of results are disputable, especially over semi-arid areas around the Mediterranean where the model tends to overestimate LAI. Over such areas, SIF is probably not proportional to GPP (the Authors assume that SIF is proportional to GPP) and the ESA-CCI soil moisture (SM) product has shortcomings (not mentioned in the current version of the paper).*

We agree that SIF has some shortcomings as GPP proxy, especially under extremely dry conditions. However, according to the study linked by the reviewer, this occurs during extreme events, and not generally over semi-arid areas:

"While a linear GPP–SIF relationship is expected in most conditions and driven by NPQ changes at the seasonal scale, under extreme stress a shift in energy allocation can occur" (Martini et al. 2022)

We propose to change l. 220 as follows:

Under extreme conditions, the linear relationship of SIF and GPP can break down (Martini et al. 2022). Therefore, similarly to FluxSat GPP, evaluations against SIF should be analysed carefully. Since we do not explicitly model SIF but only use it as GPP proxy, we evaluated it only in terms of $R$ and $R_{anom}$.

Regarding ESA-CCI soil moisture, we do not agree with the reviewer. It is true that *active* microwave-derived soil moisture has problems due to subsurface scattering in Spain, but this does not affect passive soil moisture retrievals. Since the ESA-CCI soil moisture uses a probabilistic merging approach, it takes the low reliability of the active SM retrievals in these areas into account, and assigns a relatively large weight on the passive data (see Fig. 4 in Gruber et al. 2019).

Evaluations of the ESA-CCI soil moisture product against ERA5-Land also indicates good performance in these regions: `https://qa4sm.eu/ui/validation-result/6f4b382a-60bc-4296-9b58-d8802a1b12dd`

**1.4   Incomplete analysis**

*The main problem I see in the Noah-MP DA system is that a partial (incomplete) analysis of the state variables of the soil-plant system is made. Root-zone soil moisture (RZSM) is not analyzed from the assimilation of LAI. Like LAI, RZSM changes relatively slowly. This is why RZSM needs to be analyzed together with LAI. Ignoring this tends to weaken the theoretical arguments used to criticize the bias-blind approach. The "negative effects" of the bias-blind approach are due to the incomplete use of the assimilated LAI data. Rescaling observations to the model range of values is relevant when units are different or when background model-dependent parameters affect the range of values. When model errors, model forcing errors, and model process uncertainties are responsible for the bias, the incorporation of the "true" observed values should improve the model simulations. In this case, artificially removing the bias is counterproductive and reduces the information content of the observations. This is particularly true for LAI. The exact value of this variable governs the biological regulation of soil moisture. Model LAI and RZSM biases can be due to model parameterization errors but also to biases in precipitation for example. ERA5 can present marked seasonal precipitation biases. The same difficulty would occur in irrigated areas since the Noah-MP model version used by the Authors does not represent irrigation. How can DA compensate for the impact of these biases if their influence on LAI is artificially removed? A solution is to analyze RZSM through the assimilation of the original LAI values. In the model you use, does LAI depend on RZSM?*

Thank you for this detailed and well thought out remark. We agree that a complete analysis of the soil-plant system by jointly updating LAI and RZSM might also be another worthwhile approach for handling some of the peculiarities of the current literature standard for LAI DA into Noah-MP, where normally only LAI is updated (Kumar et al. 2019; Mocko et al. 2021; Rahman et al. 2022a; Rahman et al. 2022b). The joint updates of LAI and RZSM would present an alternative solution to the problem that we tried to solve with rescaling, especially in cases where the large bias between OL and observations is caused by biases in the forcing data (e.g. seasonal precipitation biases or lack of irrigation information). We briefly mention this in lines 460-464 in our discussion:

*"[...] in strongly irrigated areas the change in soil moisture climatology leads to a decrease in soil moisture, even though the ba model performance originates from an underestimation of soil moisture due to the lack of an irrigation process in the model. Joint updates of LAI and root zone soil moisture as done in LDAS-Monde (Albergel et al., 2017) could alleviate this problem*

*caused by "missing" water to some extent but requires a good estimation of the coupling strength of LAI and soil moisture."*

We agree that this does not give the topic the attention it deserves, so we propose the following changes to the manuscript:

1. We will add an appendix section giving a more detailed overview of the Noah-MP vegetation model, and make the description of the Noah-MP vegetation model in the main text more detailed, with a focus on how parameters and RZSM can affect the equilibrium state.

2. We will add an analysis of equilibrium LAI for the Majadas pixel and the Nile delta pixel to demonstrate how the different DA setups affect the stability of the model between update steps.

In the following, we will give initial drafts for the proposed changes.

Firstly, we will add the following below line 101:

We give a more detailed overview of the dynamic leaf model in Noah-MP in Appendix B.

In the appendix, we will add the following short overview of the Noah-MP leaf model:

**Appendix B: Noah-MP dynamic leaf model**

This section gives a short overview of the Noah-MP vegetation model, focusing on the interaction of LAI and soil moisture. For a more detailed description we refer to Niu et al. (2011).

Noah-MP calculates LAI from a prognostic leaf biomass $C_l$ and a vegetation-type specific leaf area per leaf mass (specific leaf area; SLA):

$$LAI = \text{SLA} \cdot C_l$$

Leaf biomass is updated in each step via a mass balance equation:

$$\frac{\mathrm{d}C_l}{\mathrm{d}t} = \frac{1}{\text{SLA}}\frac{\mathrm{d}LAI}{\mathrm{d}t} = (1 - \text{FRAGR}) \cdot [f_l(LAI) \cdot GPP(LAI, \beta, T_c, F) - R_m(LAI, \beta, T_c)]$$
$$- D_c(LAI, T_c) - D_d(LAI, \beta) - T_l(LAI) \tag{1}$$

with FRAGR the fraction of GPP minus maintenance respiration invested in growth respiration, $R_m$ the maintenance respiration, $D_c$ the death rate due to cold stress, $D_d$ the death rate due to drought stress, and $T_l$ the turnover rate, $F$ the atmospheric forcings, and $T_c$ the canopy temperature. $f_l$ is the fraction leaf carbon allocation, which governs how much of the total assimilated carbon (GPP) is allocated to the leaf pool (the rest will be allocated to other carbon pools) and is given by

$$f_l(LAI) = \left(1 - \frac{LAI}{10}\right) \cdot \exp\left(0.01 \cdot LAI \left(1 - e^{0.75 \cdot LAI}\right)\right)$$

The dependence of vegetation growth on available soil moisture is controlled via the soil moisture factor $\beta$, which represents the relative amount of plant available water in the root zone:

$$\beta = \sum_i \frac{\Delta z_i}{z_{root}} \min\left(1, \frac{\theta_i - \theta_{wilt}}{\theta_{fc} - \theta_{wilt}}\right)$$

where $\theta_i$ is the volumetric soil moisture in layer $i$, $\theta_{fc}$ is the field capacity, and $\theta_{wilt}$ is the wilting point.

The sink terms of Equation 1 are calculated the following way:

$$R_m(LAI, \beta, T_c) = 12 \times 10^{-6} \cdot R_{25} \cdot \text{FNF} \cdot 2^{\frac{T_c - 298.16}{10}} \cdot \beta \cdot LAI \tag{2}$$

$$D_c(LAI, T_c) = 10^{-6} \cdot \frac{LAI^2}{120 \cdot \text{SLA}^2} \cdot c_c \exp\left(-0.3 \max(0, T_c - T_{c,min})\right) \tag{3}$$

$$D_d(LAI, \beta) = 10^{-6} \cdot \frac{LAI}{\text{SLA}} \cdot c_d \cdot \exp\left(-100\beta\right) \tag{4}$$

$$T_l(LAI) = 5 \times 10^{-7} \cdot c_t \cdot \frac{LAI}{\text{SLA}} \tag{5}$$

with land cover specific parameters $R_{25}$, FNF, $c_c$, $T_{c,min}$, $c_d$, $c_t$.

We will also add the following as subsection 2.6, moving the current subsection 2.6 to 2.7.

**2.6 Analysis of Noah-MP equilibrium LAI**

As we will see later, each update step in the bias-blind DA is followed by a strong drift of the model LAI towards the OL values. This indicates that there is a stable equilibrium LAI to which the model tries to return after being updated by the DA step. Calculating this equilibrium value exactly is difficult, because it depends on the current forcing conditions, model state, and model parameters, but it is possible to approximate it for typical conditions within a month.

Assuming that vegetation growth is not light limited (reasonable for the areas with large bias in the southern half of the domain), GPP is proportional to LAI and to the relative amount of plant available water $\beta$:

$$GPP(LAI) \approx \alpha(F) \cdot \beta \cdot LAI \tag{6}$$

The proportionality factor $\alpha$ depends on the set of forcings $F$, and therefore (i) varies in time and (ii) an estimation of GPP requires knowledge of all forcings. Since the forcings, and therefore also $\alpha$ have a strong seasonal component, we can simplify the analysis by making a climatological approximation, i.e., approximating $\alpha(F)$ with the typical conditions in a given calendar month $m$. This is done by relating the model-simulated GPP with model-simulated LAI and $\beta$ for the calendar month $m$, and then solving for $\alpha$ by taking the mean of $GPP/(LAI \cdot \beta)$.

We can then set the change in LAI in Equation 1 to zero and solve for the equilibrium LAI for this month, $LAI_{eq,m}$, by solving the following equation:

$$
\begin{aligned}
0 = &\beta_m \cdot (1 - \text{FRAGR}) \cdot [f_l(LAI_{eq,m}) \cdot \alpha_m \cdot LAI_{eq,m} - R_{m,w}(LAI_{eq,m}, T_{c,m})] \\
&- D_c(LAI_{eq,m}, T_{c,m}) - D_d(LAI_{eq,m}, \beta) - T_l(LAI_{eq,m})
\end{aligned} \tag{7}
$$

where $T_{c,m}$ is the mean canopy temperature and $\beta_m$ the mean plant available water for month $m$, respectively. The solution can be obtained numerically with common root finding algorithms.

We will use this approximation method also to estimate how the equilibrium LAI would change, if we would modify the model parameters or the RZSM in the model.

Based on this method section, we will also add a new figure to the results section, showing the results of the equilibrium analysis:

**3.6 Analysis of equilibrium LAI**

[Figure]

Figure 1: Estimates of the Noah-MP LAI change (dLAI/dt) and equilibrium LAI for the Majadas pixel in June (upper row), and the Nile delta pixel in July (lower row), as function of relative amount of plant available water $\beta$ (left column) and specific leaf area (SLA, right column). Additionally, the mean conditions for the OL and the DA runs are shown as dots/crosses, and the mean CGLS LAI for the respective month as line.

We additionally assessed the dependency of the equilibrium LAI value on the model root-zone soil moisture (via the relative amount of plant available water $\beta$) and on model parameters (using the specific leaf area SLA as example parameter) for the two example sites discussed above.

Figure 1 shows how the equilibrium LAI would change if we would change $\beta$ or SLA while keeping everything else constant. For both shown sites (Majadas, Nile delta) we chose the month where the mean difference between OL and observations is largest (June for Majadas, July for Nile delta). We approximated the GPP-LAI relationship for these sites and months based on Equation 6 (see Supplement for plots showing quality of fit).

For all 4 considered cases, the mean OL conditions are close to the estimated equilibrium LAI, giving confidence in the approximations performed in the derivation of the method.

[revised manuscript text omitted]

*Is RZSM analyzed when you assimilate LAI? This is not clear in the present version of the manuscript and should be made clear.*

RZSM is not analyzed jointly with LAI in our DA setup, following earlier research on LAI DA with Noah-MP, as detailed above. We apologize for not making this clear in the method section and propose to add the following sentence after l. 135

Following prior work on LAI DA with Noah-MP (Kumar et al. 2019; Mocko et al. 2021; Rahman et al. 2022a; Rahman et al. 2022b), we use the EnKF to update the model LAI, i.e., the state vector consists only of LAI.

**3    Particular comments**

**3.1**

*L. 80 (Noah-MP): More should be said here on the representation of phenology and LAI in the version of Noah-MP used by the Authors. For example, is the day-to-day change in LAI impacted by RZSM? If yes, RZSM could be analyzed through the assimilation of LAI. Is this done? If not, conclusions are only valid for this model and DA system and have no universal significance.*

As mentioned above, LAI is impacted by RZSM in Noah-MP. The detailed description is shown in the subsection 1.4.

**3.2**

*L. 120 (CGLS LAI): this product has several versions/options. Which one is used in this study?*

We used product version 2 (1 km resolution). We will clarify this in the revised version of the manuscript.

**3.3**

*L. 130-135: "land surface state" is too vague. What are the analyzed variables? Is RZSM analyzed? Please list the analyzed variables.*

Only LAI is analyzed. We will clarify this in the revised version, as proposed above.

**3.4**

*L. 223: I guess that another reason for not using RMSD is that you do not simulate SIF. Correct? Please clarify.*

Correct. We will clarify this in the revised version, as proposed above.

**3.5**

*L. 231 (ESA-CCI SM): For which soil layer? Is it surface soil moisture? Please clarify.*

We compare the ESA CCI SM product to the uppermost soil layer in Noah-MP (0-10cm).

**3.6**

*L. 267 (temperature): Do you mean accumulated precipitation?*

Yes, thanks you, this will be updated.

**3.7**

*L. 283 (section 2.6): This is a bit obscure. Probably not that interesting for a majority of potential readers. I suggest moving this part and the corresponding results to a Supplement.*

We agree with the reviewer that this section is difficult to understand, and that we might not have used the best way to explain and visualize our intention with this analysis.

The goal of this analysis is to show that the biased LAI updates, which are followed by an drift of the model LAI back towards the original equilibrium LAI, lead to similar drifts in flux estimates (GPP, ET). Furthermore, we want to demonstrate that these drifts are unphysical, i.e. they are caused by model instability instead of forcing input.

We will revise this section to make our intentions and the analysis more clear, and similarly adapt the related results section.

Section 2.6 will be revised as follows (as section 2.7)

**2.7   Evaluation of short-term DA effects**

To evaluate how biased updates affect the short-term model performance, we analyse day-to-day differences of model states and fluxes. In the OL, the day-to-day differences are driven by day-to-day variations in the forcing input. If averaged over larger areas or multiple years, this corresponds mainly to the day-to-day differences of the mean seasonal cycle. For LAI, GPP, and ET, which are high in summer and low in winter, we therefore expect positive day-to-day differences in spring, corresponding to leaf growth and increase in GPP and ET, and negative day-to-day differences in autumn, corresponding to leaf shedding and decrease in GPP and ET.

Large update steps in the bias-blind DA can induce model instabilities. In this case, the subsequent day-to-day differences are strongly impacted by the unstable artificial response to the update step, instead of reacting to the physical forcing input.

To detect if such model instabilities occur, and to what extent they propagate to flux estimates of the model, we evaluated day-to-day differences of 2 days minus 1 day after the DA update, as well as 1 day before minus 2 days before the DA update (the latter can also be interpreted as approximately 9 days minus 8 days after the DA update). A comparison of these also gives an indication of how long the DA induced effects remain.

For each pixel and month, we calculated the median of these day-to-day differences over the years 2003-2019 and normalised it with the monthly standard deviation of the variable values over the same multi-year time range (as a measure of the local within-month variation).

Additionally we will revise section 3.6 (new section 3.7) as follows

**3.7 Evaluation of short-term DA effects**

[Figure]

Figure 2: Normalised monthly median day-to-day forecast differences for (a) LAI, (b) GPP, and (c) ET. The differences are computed as the forecast value at 2-day after DA (not applied for OL) minus that of 1-day after DA (directly after DA, solid lines) and 1-day before DA minus 2-day before DA, corresponding to approximately 9-day after DA minus 8-day after DA (9-day after DA, dashed lines) for the OL (black), the bias-blind DA (blue), the CDF-matched DA (orange) and the seasonally scaled DA (green). The median was calculated from all grid cells at which the relative LAI difference between OL and bias blind DA (see Figure 1) is below -25%. For each grid cell and month, the median was normalised with the monthly standard deviation of the variable for this grid cell. The graph shows the median results across 17 years (2003-2019).

Figure 2 shows the monthly median day-to-day forecast differences for all performed simulation runs for LAI, GPP, and ET.

The OL shows a seasonal cycle with high values in spring and low values in summer, as expected (corresponds to the derivative of seasonal cycle of variable values). The bias-aware DA runs follow the OL seasonal cycle closely.

The bias-blind DA also shows the same seasonal cycle, but has an offset compared to the OL. For LAI, this offset is of the same size as the magnitude of the mean seasonal cycle, so that the mean day-to-day differences in the bias-blind DA in summer have the same magnitude as the day-to-day differences in the OL in spring, even though physically a decrease in LAI is expected. In fact, the day-to-day differences in LAI in the bias-blind DA are always positive, meaning that LAI is

expected to increase in all seasons. This is caused by the large DA update steps in the bias-blind DA, which pull the model to an unstable state. As a consequence, model instability instead of physical forcing input governs the short-term temporal evolution of LAI in the model in between update steps. Even 9 days after the DA update, right before the next update step, day-to-day differences do not significantly change, indicating that the model instability can persist for long time periods.

The instability effect also strongly affects GPP estimates throughout all seasons, and to a lesser extent ET estimates in summer.

**3.8**

*L. 331 (Figure 4): I had a hard time understanding Fig. 4. Why are CDF and seasonal simulations missing? In the caption of Figure 4 I suggest replacing: "SIF and "scaled OL" have been rescaled to have the same maximum as "bias-blind DA"" by ""scaled SIF" and "scaled OL" correspond to rescaled SIF observations and OL simulations presenting the same maximum as "bias-blind DA", respectively".*

Thank you for the suggestion of reformulating the captions, we will add it in a revised version. We did not include CDF and seasonally scaled simulations here, because we did not want to overload the plot with even more lines, since the climatologies of the CDF-scaled and seasonally scaled DA are much closer to the OL.

**3.9**

*L. 332 (deterioration of the agreement of SIF and GPP in regions with a large bias): I disagree. Regions with large bias correspond to semi-arid areas commonly affected by droughts. SIF is not linearly correlated to GPP in all conditions. In very dry conditions, this correlation disappears. See Martini et al. (2021) for example https://doi.org/10.1111/nph.17920*

See response in subsection 1.3.

**3.10**

*L. 343 (sawtooth pattern): why should "sawtooth pattern" be considered as a problem? This is a sign that DA does its job of pulling the model closer to the observations, and that increasing the number of observations would improve the simulation.*

We agree that the abrupt changes of LAI at the time of an DA update is not problematic, and simply a result of the filter algorithm doing its job. The problem is the large drift following an update, which indicates that the filter pulls the model to an unstable state and DA updates do not "stick".

As we tried to show with sections 2.6 and 3.6, this also leads to unphysical drifts/biases in flux estimates. The revised sections (now 2.7 and 3.7) are shown above.

**3.11**

*L. 349 (GLEAM ET): Can GLEAM ET be considered as a reference dataset? Why should it be better than the simulations performed by the authors?*

We included GLEAM ET for comparison due to the lack of in situ data in the areas most strongly affected by the bias-blind DA. We agree that it does not provide direct flux observations, but

(i) it is an ET product that is forced mostly with Earth observations of meteorology and land surface states and thus as close as possible to an observation-based product, (ii) it has been extensively evaluated against other products in various benchmarking activities (Greve et al. 2014; B. Martens et al. 2016; B. Martens et al. 2017; Brecht Martens et al. 2018), and (iii) it is also commonly used for assessing DA systems (e.g., Clément Albergel et al. 2019; Bonan et al. 2020; Kumar et al. 2019; Rahman et al. 2022a; Rahman et al. 2022b).

We will add the following in l. 230:

GLEAM has been evaluated against other products in various benchmarking activities (Greve et al. 2014; B. Martens et al. 2016; B. Martens et al. 2017; Brecht Martens et al. 2018), and it is commonly used for assessing DA systems (e.g., Clément Albergel et al. 2019; Bonan et al. 2020; Kumar et al. 2019; Rahman et al. 2022a; Rahman et al. 2022b). However, since it is a model-based product, comparisons against GLEAM should be analysed carefully, and GLEAM should not be treated as a ground truth validation data set.

**3.12**

*L. 361-362: These regions are also those for which microwave derived SM is more uncertain because of subsurface scattering in dry conditions (see Wagner et al. 2022, https://doi.org/10.1016/j.rse.2022.113025 )*

See response in subsection 1.3.

**3.13**

*L. 394 (Figure 9): CDF LAI is much larger than both OL and observations from January to April. Seasonally rescaled LAI is much larger than both OL and observations from April to July 2016. How is this possible? Rescaled LAI should be somewhere between the OL and the observations. Correct?*

We are unsure how to interpret this comment: Does the reviewer (i) wonder why the seasonally rescaled LAI analysis (green line) is above both OL (black line) and original observations (blue dots), or (ii) wonder why the seasonally rescaled LAI observations (green dots) is above both OL (black line) and original observations (blue dots)?

In case (i), we want to point the reviewer to the seasonally rescaled observations (green dots), which are the assimilated observations. The seasonally rescaled analysis (green line) is between these observations and the OL, as expected. We will revise the figure caption to make the different dots and lines more clear.

If the author wonders about case (ii): The rescaling tries to transform the original observations in a way such that certain statistics of the OL are reproduced in the rescaled observations. In the case of the seasonal rescaling, we try to match the mean seasonal cycle and the overall standard deviation. It neither can nor intends to induce or conserve any ordering relationships with respect to the OL. Instead, it conserves the ordering relationships in the original observations. For example, if in the original observations April 2016 was the April with the highest LAI values within the baseline period (2002-2019), April 2016 will also be the April with the highest LAI values within this period in the rescaled observations, no matter what the OL values are.

**3.14**

*L. 399: replace "suppressing" by "reducing".*

Thank you, we will update the manuscript accordingly.

**3.15**

*L. 401 (Figure 10): Why is the number of curves/dots in Fig. 10a different from Fig 9a? This not logical.*

In the rescaled DA runs not much is happening in this example. We therefore excluded the lines to not overload the plot with irrelevant lines. We can also include the lines or add the complete plot in the supplements, if required.

**3.16**

*L. 404 (strongly decreases SM2): is this because RZSM is not analyzed?*

The reason for this is that the increase in LAI also strongly increases transpiration, which in turn decreases RZSM/SM2. Updating also RZSM in this case would mean adding water. If the system is properly tuned and cross-correlations are estimated well, this might counterbalance the increased water uptake. See subsection 1.4 (this document) for a more in depth discussion of the question of joint RZSM+LAI updates.

**3.17**

*L. 409 (section 3.6): Move this section to a Supplement.*

We refer to previous comments on this issue.

**3.18**

*L. 440-442: ... and possible seasonal biases in ERA5 precipitation*

Thank you, we will add this in a revised section of the manuscript.

**3.19**

*L. 494: For the sake of clarity, it should be written here that rescaling LAI observations has a negative impact on DA efficiency.*

We rephrased the conclusion to make this more clear.

**3.20**

*L. 496: Are "standard assumptions" valid in a context where key variables (such as RZSM in this study) are not analyzed?*

Even in such a scenario, the diagnostics based on these assumptions can be used to assess if the Kalman filter update sequence has the intended statistical properties, i.e. if the standard assumptions are valid.

**3.21**

*L. 521: Do you mean that RZSM has no impact on the simulated LAI? This is far from the state of the art. Is there a more advanced version of Noah-MP able to correctly simulate LAI?*

RZSM impacts LAI in Noah-MP, as elaborated in more detail above. Only the phenology index (growing season vs. no growing season) does not take into account water-driven phenology.

**3.22**

*L. 550: I completely disagree with this recommendation. I would instead recommend paying attention to the consistency between LAI and RZSM in the LSM, and to the "fitness for purpose" of the Noah-MP LSM.*

See comment in subsection 1.4.

---

## Author Comment (AC2)

**Reply to Comment by Anonymous Referee #2**

April 3, 2023

We thank the anonymous reviewer for the helpful and detailed comment.

In the following, review comments will be styled in *blue italics*, our responses in black normal text, and proposed changes in green.

**1.1**

*I think both the abstract and conclusion do not fairly reflect the results and discussion sections. Benefits and pitfalls have been discussed for both bias-blind and bias-aware approach and the bias-blind approach leads to greater improvements for most of the variables and most of the metrics. However, the abstract/conclusion leads to the recommendation of the bias-aware approach, which is partly inconsistent with the message sent from the result and discussion section. I would recommend the authors to reconsider making the key points that can better and fairly reflect the content.*

In our conclusion we tried to point out the specific circumstances under which the bias-aware DA might have benefits over the current literature standard of bias-blind DA. Based on your and the other reviewers feedback we agree that this can give a false impression of disregarding the bias-blind DA. We will therefore add the following paragraph after l.549:

The bias-blind DA is most effective at reducing the disagreement in modelled and observed LAI, and leads to the largest improvements in GPP and runoff. For many applications it is therefore a suitable option, especially if large bias reductions are intended, even if bias-blind Kalman filtering is suboptimal. A temporal interpolation of the observation data, or even a direct insertion approach, could be even more efficient in this case. However, this approach does not necessarily improve other variables, e.g. if the model simulates biased LAI in conjunction with unbiased soil moisture. As alternative, we recommend to use observation rescaling techniques for LAI DA with Noah-MP if there are strong biases and if

- ... (same as in original manuscript)

Additionally, we will reformulate the last paragraph of the conclusion as follows:

A drawback of the observation rescaling approaches is that they result in estimates in the model climatology. If the observation-forecast bias is due to erroneous precipitation forcing or missing irrigation input, joint updates of LAI and RZSM in a bias-blind system can be considered instead. This might lead to large bias corrections while still retaining a stable model state even after large updates. However, if the bias is not only caused by bias in the precipitation/irrigation, this poses the risk of seriously degrading the soil moisture estimates.

Alternatively, updates to model parameters, either via joint parameter and state update DA, or via a priori model calibration can also lead to more stable and persistent updates and LAI estimates in the observational climatology. This is especially desirable for research on the carbon cycle, where absolute values of carbon fluxes are required. Parameters to consider for calibration are parameters related to model leaf growth, but potentially also photosynthesis or soil parameters.

To gain the most benefit from LAI data assimilation into Noah-MP, further research and model improvement of the coupling mechanisms between water and carbon cycle is necessary.

**1.2**

*L48-49: "It is possible that other processes (e.g., transpiration) are only represented well for a biased model climatology". If that turns out to be true, isn't it right for the wrong reason? I think such side effect should be fixed by improving the model physics instead of regarding as the weakness in the "bias-blind" DA approach.*

We agree that the best way to fix such effects would be by improving the model physics. However, this is out of scope of this study, where we only consider updates to the LAI state in the model. We discuss parameter update DA or model calibration as alternatives to updating only the LAI state, which might achieve the same effect as improving the model physics.

**1.3**

*The spatial resolution of the simulation is coarse while there are evaluation reference datasets from in situ observations. The scale mismatch is not well considered in the metrics and comparisons in this paper, which may provide biased assessments. Would it be possible to conduct the simulation at a finer spatial resolution if most of the input datasets and the LAI observations are available at finer scale?*

We used the ERA-5 reanalysis as forcing data, which are not available with a finer spatial resolution. We therefore do not believe that a simulation at a finer spatial resolution will significantly improve the model results, at least in terms of anomaly-based metrics, while it would at the same time strongly increase our computational demands.

Due to the scale mismatch, we limit the comparisons of in situ soil moisture data to anomaly-based comparisons, as mentioned in l. 249-250 in the original manuscript:

*"Since soil moisture climatology and absolute values strongly depend on sub-grid scale factors like slope and soil texture, we only compared the in situ values in terms of anomaly correlation $R_{anom}$."*

Additionally, we mentioned in our discussion that the scale mismatch hampers our assessment, l. 467-470: *"This can be due to assumptions and errors in the underlying satellite data and retrieval algorithms in the case of satellite-based data, or due to different spatial support in the case of in situ data. Hence, whether the bias-blind DA leads to estimates closer to the "truth" remains uncertain, and evaluations with different reference products might come to different conclusions"*

To increase the reliability of the comparisons, we propose to restrict the evaluation to ISMN stations that have been shown to be representative at the coarse scale in a triple collocation analysis including ISMN, ERA5-Land volumetric soil moisture layer 1, and ESA CCI soil moisture.

**1.4**

*L102 - 105: What is the spatial resolution of these input datasets as well as the ERA5 forcing datasets? Please clarify in the text.*

The parameter datasets are available on a 0.01° regular grid. The ERA5 forcing dataset has an original resolution of 31 km and has been rescaled to a 0.25° regular grid. We will add the following sentence in l. 114:

The soil texture and land cover maps are available on a 0.01° regular grid and have been upscaled to a 0.25° grid using the largest fraction within a model grid cell. The ERA5 forcings are have an original resolution of 31 km and have been rescaled to a 0.25° regular grid.

**1.5**

*L106-108: I didn't quite understand why the model interprets the evergreen broadleaf forests as tropical rainforests. And what land cover type does the UMD data assign to for these locations? Are there any references that can justify the substitution of UMD data is more realistic?*

The Noah-MP vegetation model allocates GPP to leafs according to a parametric allocation function. This function makes a distinction between evergreen broadleaf forests (EBF) and all other land cover types such that the maximum LAI is approximately 8 for EBF and approximately 6 for all other land cover types (see Figure 1 in this document). This is done to account for the much higher productivity of tropical rain forests than other land cover types. However, we found that for the pixels in Europe assigned to EBF this approach produced even more strongly biased LAI estimates than we showed in our manuscript. Replacing the land cover of these pixels with UMD as fallback option reduced this bias.

[Figure]

Figure 1: Allocation of total photosynthesis (GPP) to leaf mass balance

**1.6**

*L116: Is there a specific reason to use a spatial resolution of 0.25 degree? Why not bring it to a finer resolution since your CGLS LAI product is at such a fine resolution.*

As mentioned above, a simulation on a much finer scale would be a lot more computationally demanding, while at the same time the benefit is questionable, since the forcing data is only available on a coarse resolution.

**1.7**

*L125: Is there a particular reason not using the temporal interpolation method? If this can better deal with the sawtooth issue, why not apply it?*

We do believe that the interpolation technique might be a viable solution for some of the issues associated with the bias-blind approach, but for this study we had 3 particular reasons not to perform interpolation:

- Interpolation can only be reasonably used if the assimilated LAI product is smooth enough, so that we can assume that the pseudo-observations introduced by the interpolation are close to the true observations. This is the case for the CGLS LAI product, but not for other products. By not interpolating the data, we firstly stay closer to how the data is provided, and secondly we also obtain more insights into how observation frequency (and possibly changes therein) affect the data assimilation. This could be especially useful if microwave vegetation optical depth is used as LAI proxy, similar to Mucia et al. (2021).

- Interpolation of data will introduce a strong auto-correlation of observation errors. However, our data assimilation framework assumes that observation errors are uncorrelated.

- The main effect that interpolation has is to pull the model closer to the observations and hide the drifts between DA updates better. If this is the intended goal, a direct insertion technique might be even more suitable than a Kalman filter.

To make our reasoning more transparent, we will modify l.125 as follows:

In contrast to Kumar et al. (2019b), we did not interpolate the LAI to daily values. This way we (i) do not introduce observation error auto-correlations, (ii) allow our results to be generalizable to LAI data sets (or proxy data sets as used in S. V. Kumar et al. (2020) and Mucia et al. (2021)) with less frequent observations or changes in observation frequency, and (iii) can investigate if the filter efficiently interpolates and operates as intended (or assumed). We assimilated the aggregated data every 10 days at 0:00 UTC, where and when they are available.

Additionally, we will add the following paragraph in the discussion, after l. 485:

The sawtooth pattern can be reduced by interpolating the observations or applying time series smoothing methods to obtain pseudo-observations at a daily frequency. This will keep the analysis closer to the observations and prevent model drift over multiple days. However, in this case direct insertion approaches or using observed LAI directly as model parameter could achieve even better results than an EnKF.

**1.8**

*L163: For the seasonal scaling approach, how does the phase of seasonality look like between the LAI observation and the model simulation for the study domain? For instance, what does the spatial map of the peak month in LAI observation compared to the OL simulation? The vegetation scheme in Noah-MP has weakness even in reasonably estimating the magnitude and phase of the seasonal cycle of vegetation growth. It may introduce additional bias if rescaling observation based on the modelled climatology. Any comments on this?*

The bias-aware LAI DA will lead to LAI estimates in the model climatology, so in fact any biases in magnitude and phase of the seasonal cycle in the model will, by design, still be present in the analysis. We agree that estimates in the observation climatology, as obtained with the bias-blind approach, might be more suitable for some applications, but this comes with the drawbacks

presented in our study (model drift, biased flux estimates, large side effects). As discussed in the manuscript, a better calibration of model parameters might combine the respective advantages of the two approaches. In the revised section, we will also add a discussion of joint LAI and RZSM updates as further alternative, which could also help to obtain stable estimates in the observation climatology.

**1.9**

*L247: Considering the coarse spatial resolution of the model set up, the scale mismatch much be an outstanding issue when comparing the gridded soil moisture value to the in-situ observations. A simple nearest neighbor matching between ISMN stations and model grid might be troublesome. Have the authors considered the representativeness of the ISMN data for a model grid? Please comment on it and potentially discuss the uncertainties. I think simulation performed on a much finer scale might be better if one were to directly compare the in-situ observation to simulated values for a model grid.*

Comparisons to in situ soil moisture are commonly done also for coarse model simulations, Sujay V. Kumar et al. (e.g., 2014), Sujay V Kumar et al. (2019), Albergel et al. (2020), Rahman et al. (2022a), Rahman et al. (2022b), and Heyvaert et al. (2023, in review). To account for bias due to the different spatial support/representativeness, e.g. due to different soil texture we only compare soil moisture anomalies.

Additionally, as mentioned above, we propose to restrict our analysis to ISMN stations considered representative at a coarse scale.

**1.10**

*L260 -262: Why not use the dam and irrigation specific datasets to mask out the basins that are heavily affected by these factors? I think directly masking out the basins based on the correlation threshold of 0.4 is biased because it is impossible to justify that such low correlation is surely due to the unmodeled processes. Such selective results may mislead audience in terms of the DA performance. I would recommend the authors reconsider the approach.*

This threshold removes only very few outlier stations, while most other stations have correlation values larger than 0.6. For reference, we provide the original correlations of all considered stations in Figure 2 (this document).

**1.11**

*L308 & L402 -403: I wonder if the case of Nile delta may not simply because of the lack of irrigation representation in the model. How does the LAI time series look like for the full study period compared to the time series of precipitation? Did you see consistently low LAI regardless of dry and wet years? I wonder if the representation of vegetation in response to water stress factor or the mismatch of soil types may play a role in the dynamic vegetation model that limits the growth of vegetation for this area? There are multiple reasons that the model may under-estimate/overestimate vegetation growth just as the author discussed in section 4.4, I strongly recommend the authors take a closer look at the data and elaborate more on the discussion in terms of the factors at play. The analysis presented here does not justify that the lack of irrigation forcing is the main reason.*

OL LAI is consistently lower than the observations, but shows strong interannual variability

[Figure]

Figure 2:   Correlation of GRDC runoff with OL.

related to water availability. An additional analysis of the sensitivity of the model equilibrium LAI on RZSM showed that in the Nile delta LAI is highly sensitive to RZSM.

We will add the following paragraph in the discussion:

In the Nile delta, a known source of bias is the missing irrigation input in the model. Additionally, we found a very strong water limitation on vegetation growth, and a strong sensitivity of the equilibrium LAI on changes in RZSM, implying a strong model coupling of LAI and RZSM. Therefore, joint updates of LAI and RZSM are likely to improve the LAI DA results here, because RZSM is temporarily adjusted, but changes to soil parameters might also be necessary to sustain the increased moisture values.

**1.12**

*L315 - 317: This explains places where bias-blind DA leads to LAI decrease and soil moisture increase. I wonder how would the authors interpret places where bias-blind DA indicates an LAI increase, while there is no soil moisture decrease?*

The areas where the bias-blind DA increases growing-season LAI but does not decrease soil moisture are high-altitude regions in the Alps and Norway. These areas have some specific attributes that limit the impact of LAI change on soil moisture:

- Precipitation is frequent and keeps the soil moisture high. Any decrease in soil moisture would quickly be refilled. The same happens during the rainy season at the Majadas pixel, see Figure 9 in the manuscript.

- These mountaineous areas have steep slopes, the surface water balance is therefore dominated by runoff and ET has a relatively lower role.

- The areas are cold and therefore feature a low atmospheric water demand. ET therefore is also low, especially compared to runoff.

We will add the following in l. 319:

In areas where growing-season LAI is decreased by the bias-blind DA the response differs. In the Nile delta, the increase in LAI leads to a reduction in soil moisture via transpiration. In the Alps and Scandinavia, soil moisture is not affected systematically, since the water balance is dominated by runoff, and transpiration changes therefore have a relatively lower impact.

**1.13**

*L324-325: The anomaly correlation of GPP improves a lot for places where LAI bias is large. Isn't it persuasive that the raw LAI observation plays an essential role in constraining the inter-annual changes in vegetation growth? The bias-aware approach may limit such benefit.*

Agreed. As indicated above, we will reformulate our conclusions to point this out.

**1.14**

*L334: It is not clear to me what area has been covered in the "southern part of the domain". I would suggest mark it out in Figure 3 or simply provide a mask map along with Figure 4 to highlight area of analysis.*

This refers to "the high-bias regions in the southern part of the domain", which we define as "all model grid cells south of 42° where the relative LAI difference is lower than -30%" (see caption to Figure 4 in the manuscript). We can add a figure marking these areas in a supplement.

**1.15**

*ET comparison: Considering large uncertainty in ET products, would it be valuable to include a few other ET datasets for comparison besides GLEAM ET?*

We agree that more reference data sets are always better, but (i) we did not want to extend the already quite long manuscript, and (ii) the amount of information that can be obtained from the comparisons to satellite or model derived ET datasets is limited, especially if we want to evaluate the effects of bias, since the bias of all of these with respect to the truth is unknown. Since changes in ET are lower than in GPP, we decided to do a more detailed evaluation with GPP instead. If deemed useful by the editor, we can include a more detailed comparison.

**1.16**

*Figure 9: The seasonal cycle of the LAI observation would be very different if scaled by CDF or seasonal factors. I highly doubt whether such scaling reduces the value of using LAI observation to constrain the model performance. Scaling the observation to a model climatology that is further away from the truth is another form of bias. This might explain partly that the two bias-aware DA does not lead to better improvements for GPP as compared to the bias-blind DA.*

We fully agree with these sentiments, and will adapt the discussion & conclusion as detailed above.

**1.17**

*L387: About the Majadas site: I'd suggest informing where the site is located in Figure 5.*

We will update Figure 5 with locations of the Majadas site and the selected Nile delta example pixel.

**1.18**

*L375-378: Repeated sentence. And I think such sawtooth pattern could be taken care of if enables the temporal interpolation to the observations.*

The temporal interpolation would at least visually remove the sawtooth pattern with daily output, but effects like the unphysical short-term dynamics of LAI and GPP would still be at play. However, we will also mention this in the revised discussion giving more space to the bias-blind DA.

**1.19**

*Section 3.4: Again I have concerns in the scale mismatch in the modeling space vs. the observation. Without consider such effect, the conclusion may be biased. Would it be possible to conduct a sensitivity analysis regarding the spatial resolution of the model? Alternatively, is it possible to collect more in situ observation sites that can better jointly represent the condition for a 0.25 degree grid cell?*

Unfortunately we do not have more in situ observation available, but we will instead restrict the analysis to stations that are deemed representative at the coarse scale.

**1.20**

*L405-408: Good point! How does the other two bias-aware DA perform in this case?*

We did not add the bias-aware DA results in these plots because they are very similar to the OL. If requested, we can add them here or in a supplement.

**1.21**

*L426-427: True but the seasonal scaling may introduce additional bias to the scaled observations if the modelled climatology is further away from the ground truth. I think this is probably a larger flaw as compared to the bias-blind DA approach because at least the later remains the true temporal variation of what has been observed.*

We agree that there is a trade-off between staying close to the observations and having a well-behaved filter algorithm with a stable model. We discuss this in more detail in the proposed revisions above.

**1.22**

*Section 4.2 - 4.3: I agree that if large bias exists then the bias-blind DA may lead to misuse of Kalman filter; but I also think that the rescaling of observation based on a biased model is a big issue, it may create even bigger problem if the seasonality is deteriorated. I think the benefit and pitfalls for both approaches should be elaborated and discussed in a more comprehensive manner before it gets to section 4.4.*

We again refer to the changes to the discussion and conclusion mentioned above.

---

## Author Response (AR2)

**Reply to Comment by Anonymous Referee #2**

S. Scherrer, G. De Lannoy, Z. Heyvaert, M. Bechtold, C. Albergel ,T. S. El-Madany, W. Dorigo

August 23, 2023

We thank the anonymous reviewer for the recommendations and adapted the manuscript accordingly.

In the following, review comments will be styled in *blue italics*, our responses in black normal text, and proposed changes in green.

**1**

*Regarding the title, it might be useful to consider that some readers may not be familiar with the term "Noah-MP". To enhance clarity and avoid potential ambiguity, I recommend revising it to "Noah-MP Land Surface Model".*

We agree, the revised title is Bias-blind and bias-aware assimilation of leaf area index into the Noah-MP land surface model over Europe.

**2**

*While I commend the authors' approach to presenting the DA impact on LAI equilibrium and suggesting for multi-DA or parameter updates for reducing model instability, the rational for this analysis isn't immediately clear within section 2.6. To enhance understanding, it would be beneficial to 1) clarify the purpose of this analysis at the beginning; 2) defining LAI equilibrium and its relationship with soil moisture; 3) simplify and reduce the technical details of the approximation method (move it to appendix) or clearly states why it is used and how it defers from the model physics; and 4) perhaps add more context before bringing up equation 2 or move it to Appendix as well and provide with a simpler version here.*

We agree that the motivation for this analysis was not clear immediately in the original version. We rephrased the first paragraph of section 2.6 as follows:

As will be seen later, each update step in the bias-blind DA is followed by a strong drift of the model LAI towards the earlier forecast values, i.e. the bias-blind DA system quickly "forgets" systematic corrections made in earlier steps. This indicates that there is a stable equilibrium LAI (i.e. a model-based 'attractor') whose value is not modified by the bias-blind LAI DA. To make full use of the information contained in the observations, a bias-blind DA system should also modify this equilibrium LAI value to have more persistent DA updates.

Additionally, we followed the recommendation to move the technical details, including the defining equation of the equilibrium LAI to the appendix, and only shortly summarise the results in section 2.6 as such:

An analysis of the Noah-MP leaf growth model (Appendix A) shows that the main factors influencing the equilibrium LAI value are (i) root zone soil moisture, represented via the soil moisture factor $\beta$, and (ii) leaf parameters, e.g. specific leaf area (SLA, leaf mass per area). Including $\beta$ or SLA in the DA state vector could thus help to obtain more persistent updates.

We therefore analysed how sensitive the equilibrium LAI is to these variables using a climatological approximation of the Noah-MP leaf model (shown in Appendix B). The result of this analysis is presented in subsection 3.6 for two example sites with constrasting bias between Noah-MP and CGLS, (i) the Majadas site in Spain, where observed LAI is much lower than modelled LAI, and (ii) the Nile delta, where observed LAI is much higher than modelled LAI.

In the appendix, we added the following section that gives more details on how we found the climatological approximation for the model physics:

The equilibrium LAI value (model-based 'attractor') is the LAI value at which Equation A1 is zero. It is therefore an implicit function of all the terms and variables on the right hand side of Equation A1. However, some of the terms on the right hand side of this equation strongly depend on the meteorological forcings (e.g., GPP). Evaluating the equilibrium LAI as function of soil moisture and leaf parameters would therefore require running the complete Noah-MP model with a wide range of forcing conditions, which quickly becomes computationally intractable. Therefore, we use a climatological approximation of the term in Equation A1 to eliminate the explicit dependence on the meteorological forcings.

We obtain a climatological approximation of GPP as a function of LAI and $\beta$ by assuming that GPP is proportional to LAI, $\beta$, and a factor that depends solely on the forcings $F$ or constant parameters not including the leaf parameters:

$$GPP(LAI, \beta) \approx LAI \cdot \beta \cdot \alpha(F) \tag{1}$$

The linear dependence on $\beta$ is part of the Noah-MP model physics, while the assumption of linear dependence on LAI is justified in case vegetation growth is not light limited. This is reasonable for the areas with a large bias in the southern part of the domain, where vegetation growth is mainly water limited. To find an approximation for $\alpha$, we perform a least-squares fit of Equation 1 using daily mean model output for GPP, LAI, and $\beta$ for each calendar month. This results in 12 separate approximations of $GPP(LAI, \beta)$, one for each calendar month. The fit for the month with the highest discrepancy between Noah-MP OL and CGLS are shown in Figure 1

For the other terms in [Equation A1 in the manuscript], we simply insert the mean forcing value if required. The resulting defining equation for the equilibrium LAI for the month $m$ is then

$$
\begin{aligned}
0 = &\beta_m \cdot (1 - \text{FRAGR}) \cdot [f_l(LAI_{eq,m}) \cdot \alpha_m \cdot LAI_{eq,m} - R_{m,w}(LAI_{eq,m}, T_{c,m})] \\
&- D_c(LAI_{eq,m}, T_{c,m}) - D_d(LAI_{eq,m}, \beta) - T_l(LAI_{eq,m})
\end{aligned} \tag{2}
$$

where $T_{c,m}$ is the mean canopy temperature, $\beta_m$ the mean plant available water, and $\alpha_m$ the GPP proportionality factor from Equation 1 for month $m$, respectively. The solution can be obtained numerically with common root-finding algorithms.

[Figure]

Figure 1: [Figure B1 in manuscript] Dependence of soil-moisture-normalized GPP (GPP/$\beta$) on LAI for the OL and DA runs, and linear approximation via least-squares fit for (a) Majadas in June and (b) the Nile delta in July.

**3**

*Regarding section 3.3, if the decision is to not incorporate additional reference ET datasets for comparison, it would be beneficial for the authors to extend their discussion, specifically addressing the potential shortcomings or uncertainties of GLEAM ET datasets for specific regions, such as those under irrigation agriculture. Besides, I think providing references to literature that addresses ET uncertainties or comparisons within the study region could also strengthen this section. Given that the bias-blind DA has shown large impact (both positive and negative) on ET compared to bias-aware DA, it is crucial to provide a more comprehensive analysis or discussion surrounding these effects. This becomes especially necessary considering the current lack of consensus regarding a state-of-art reference ET product.*

We rephrased section 2.5.3 and added a paragraph to highlight the shortcomings of GLEAM for strongly irrigated areas:

The Global Land Evaporation Amsterdam Model v3 (GLEAM; Martens et al., 2017; Miralles et al., 2011) ET dataset is a gridded ET product based on a land surface model and satellite observations. It has been evaluated against other products in various benchmarking activities (Greve et al., 2014; Martens et al., 2016, 2017, 2018), and it has been used for assessing DA systems (e.g., Albergel et al., 2019; Bonan et al., 2020; Kumar et al., 2019b; Rahman et al., 2022b, a). We used version 3.6b, as it provides data in our evaluation period (2003-2019) and does not rely on either reanalysis as forcing data or optical data for dynamic inputs. It is thus largely independent of the assimilated CGLS LAI and of the Noah-MP-modelled ET, but inevitably suffers from model assumptions and input errors.

GLEAM calculates ET as a combination of potential evaporation (based on the Priestley-Taylor equation), stress, and interception (based on the Gash model). Water stress is based on a soil moisture model included in GLEAM, and an additional scaling based on observations of vegetation optical depth, a proxy for vegetation water content.

Since the soil moisture model does not include irrigation explicitly, it will provide biased estimates over strongly irrigated areas (Chen et al., 2021; Shah et al., 2019). Evaluations of absolute values (e.g., via RMSD) over irrigated areas should therefore be analysed carefully, as they might show decreased performance stemming from an actually improved representation of irrigation (as for example in Thiery et al., 2017), even if satellite-based soil moisture anomalies were assimilated and might partly compensate for missed irrigation.

*Regarding Figure 6, it appears that the bias-blind approach exhibits notable improvements in anomaly correction for SM2 compared to ISMN sites. Could this potentially indicate an enhancement of deeper zone soil moisture? Why this is the case? And how does the correlation look like instead of anomaly correlation? It would be helpful if the authors could elaborate on what the soil moisture climatology looks like for these sits where marked improvements in anomaly R are observed. Additionally, is there a specific reason not including the R analysis for soil moisture evaluation? This is necessary to provide insightful information about the impact of DA on soil moisture seasonality and provide more evidence for the statement made in the abstract.*

We agree that there is some value in also providing the raw correlation values and we added them in Figure 6, along with a short description of the results. The revised figure is shown in Figure 2 (this document, Figure 6 in the manuscript).

[Figure]

Figure 2: Top row: (a) Map of NIC $R_{anom}$ with ESA CCI SM for the bias-blind DA and (b) box plots of NIC $R_{anom}$ with ESA CCI SM for all three DA runs. Middle row: Maps of NIC $R_{anom}$ with ISMN for the bias-blind DA for (c) SM1 (0-10 cm) $R_{anom}$ and (d) SM2 (10-40 cm) $R_{anom}$, and (e) box plots of NIC $R_{anom}$ with ISMN for SM1 and SM2 and all three DA runs. Bottom row: Maps of NIC $R$ with ISMN for the the bias-blind DA for (f) SM1 (0-10 cm) $R$ and (g) SM2 (10-40 cm) $R$, and (h) box plots of NIC $R$ with ISMN for SM1 and SM2 and all three DA runs.

The raw correlations show a similar pattern of improvement as the anomaly correlations, and we indeed believe that this is due to improved subsurface soil moisture. However, as none of the ISMN stations are in the areas of strong bias, this finding is of limited value to discuss the influence of biased DA on the climatology of root zone soil moisture.

We added the following sentence in the discussion section 4.1:

The comparison to ISMN indicates improvements in deeper layer soil moisture, but none of the in situ sites considered are in the areas with large bias.

Additionally, we rephrased the sentence in the abstract concerning soil moisture climatology as follows:

While comparisons to in-situ soil moisture in areas with weak bias indicate an improvement of the representation of soil moisture climatology, bias-blind LAI DA can lead to unrealistic shifts in soil moisture climatology in areas with strong bias.